# Quantum and Quantum-Inspired Stereographic K Nearest-Neighbour Clustering

**DOI:** 10.3390/e25091361

**Published:** 2023-09-20

**Authors:** Alonso Viladomat Jasso, Ark Modi, Roberto Ferrara, Christian Deppe, Janis Nötzel, Fred Fung, Maximilian Schädler

**Affiliations:** 1Theoretical Quantum System Design Group, Chair of Theoretical Information Technology, Technical University of Munich, 80333 Munich, Germany; janis.noetzel@tum.de; 2Institute for Communications Engineering, TUM School of Computation, Information and Technology, Technical University of Munich, 80333 Munich, Germany; roberto.ferrara@tum.de (R.F.); christian.deppe@tum.de (C.D.); 3Optical and Quantum Laboratory, Munich Research Center, Huawei Technologies Düsseldorf GmbH, Riesstr. 25-C3, 80992 Munich, Germany; fred.fung@huawei.com (F.F.); maximilian.schaedler@huawei.com (M.S.)

**Keywords:** quantum k nearest-neighbour, quantum machine learning, quantum computing, *k*-means clustering, 6G communication, quadrature amplitude modulation, quantum-classical hybrid algorithms, quantum-inspired algorithms

## Abstract

Nearest-neighbour clustering is a simple yet powerful machine learning algorithm that finds natural application in the decoding of signals in classical optical-fibre communication systems. Quantum *k*-means clustering promises a speed-up over the classical *k*-means algorithm; however, it has been shown to not currently provide this speed-up for decoding optical-fibre signals due to the embedding of classical data, which introduces inaccuracies and slowdowns. Although still not achieving an exponential speed-up for NISQ implementations, this work proposes the generalised inverse stereographic projection as an improved embedding into the Bloch sphere for quantum distance estimation in k-nearest-neighbour clustering, which allows us to get closer to the classical performance. We also use the generalised inverse stereographic projection to develop an analogous classical clustering algorithm and benchmark its accuracy, runtime and convergence for decoding real-world experimental optical-fibre communication data. This proposed ‘quantum-inspired’ algorithm provides an improvement in both the accuracy and convergence rate with respect to the *k*-means algorithm. Hence, this work presents two main contributions. Firstly, we propose the general inverse stereographic projection into the Bloch sphere as a better embedding for quantum machine learning algorithms; here, we use the problem of clustering quadrature amplitude modulated optical-fibre signals as an example. Secondly, as a purely classical contribution inspired by the first contribution, we propose and benchmark the use of the general inverse stereographic projection and spherical centroid for clustering optical-fibre signals, showing that optimizing the radius yields a consistent improvement in accuracy and convergence rate.

## 1. Introduction

Quantum Machine Learning (QML), using quantum algorithms to learn quantum or classical systems, has attracted much research in recent years, with some algorithms possibly gaining an exponential speedup [1,2,3]. Since machine learning routines often push real-world limits of computing power, an exponential improvement to algorithm speed would allow for such systems with vastly greater capabilities [4]. Google’s ‘Quantum Supremacy’ experiment [5] showed that quantum computers can naturally solve specific problems with complex correlations between inputs that can be incredibly hard for traditional (“classical”) computers. Such a result suggests that machine learning models executed on quantum computers could be more effective for specific applications. It seems quite possible that quantum computing could lead to faster computation, better generalisation on less data, or both, for an appropriately designed learning model. Hence, it is of great interest to discover and model the scenarios in which such a “quantum advantage” could be achieved. A number of such “Quantum Machine Learning” algorithms are detailed in papers such as [2,6,7,8,9]. Many of these methods claim to offer exponential speedups over analogous classical algorithms. However, some significant gaps exist between theoretical prediction and implementation on the path from theory to technology. These gaps result in unforeseen technological hurdles and sometimes misconceptions, necessitating more careful case-by-case studies such as [10].

It is known from the literature that the *k*-nearest-neighbour clustering algorithm (*k*NN) can be applied to solve the problem of phase estimation in optical fibres [11,12]. A quantum version of this *k*NN has been developed in [2], promising an exponential speedup. However, the practical usefulness of this algorithm is under debate [4]. The encoding of classical data into quantum states has been proven to be a complex task which significantly reduces the advantage of known quantum machine learning algorithms [4]. There are claims that the speedup is reduced to only polynomial once the quantum version of the algorithm takes into account the time taken to prepare the necessary quantum states. Furthermore, for near-intermediate scale quantum (NISQ) [3] applications, we should not expect the availability of QRAM, as this assumes reliable memories and operations which are still several milestones out of reach [13]. For this reason, it is not currently possible to use the fully quantum clustering algorithm and thus we resort to using *hybrid quantum-classical k*NN algorithms. Any classical implementation of *k*NN clustering involves, among other steps, repeated evaluations of a dissimilarity and and a loss function; changing the dissimilarity leads to a different clustering. A hybrid quantum classical *k*NN clustering algorithm utilizes quantum methods only to estimate the dissimilarity, eliminating the need for long-lasting quantum memories. However, reproducing the dissimilarity of a classical *k*NN algorithm using quantum methods can be prohibitively restrictive. The quantum dissimilarity also depends on the embedding (how the classical data are encoded in quantum states) and might only approximate the classical one, introducing fundamental deviations from the classical *k*NN algorithm. In [10], we applied a hybrid quantum-classical algorithm with modified angle embedding to the problem of *k*-means clustering for 64-QAM (Quadrature Amplitude Modulation) optical-fibre data (a well-known technical problem in signal processing through optical-fibre communication links) provided by Huawei [14], and show that this does not currently yield an advantage due to both the embedding and the current speed and noise of quantum devices.

In this work, we use the same problem and datasets to bring two main but independent contributions using the generalised inverse stereographic projection. First, we embed classical 2-dimensional data by computing the ISP onto the 3-dimensional sphere, and use the resulting normalised vector as the Bloch vector to produce a pure quantum state of one qubit, which we call stereographic embedding. The resulting quantum dissimilarity directly translates into the cosine dissimilarity, thus making the quantum algorithm mathematically closer to the classical *k*-means algorithm. This means that no inherent limitation is introduced by the embedding and any loss in performance of this hybrid algorithm can be compensated for by improving the noise level and the speed of the quantum device. We thus propose stereographic embedding as an improved quantum embedding that may lead to improvement in several quantum machine learning algorithms (although there might not still be a practical quantum time advantage).

The second contribution comes from the benchmarking of the hybrid stereographic quantum mentioned above. Since, as already mentioned, the resulting hybrid clustering algorithm is mathematically equivalent to a classical ‘quantum-inspired’ *k*NN algorithm, in order to assess its performance in the absence of noise, we simply test the equivalent classical quantum-inspired *k*NN algorithm. This algorithm is the result of first computing the ISP of the data and then performing clustering using a novel ‘quantum’ centroid update. We observe an increase in accuracy and convergence performance over *k*-means clustering on the 2-dimensional optical-fibre data. This suggests, as a purely classical second main contribution, that an advantage in decoding 64-QAM optical-fibre data is achieved by performing clustering in the inverse stereographically projected sphere and by using the spherical centroid.

This paper is structured as follows. In the remainder of this introduction, we discuss related works and our contribution to it. In Section 2, we introduce the experimental setup generating the 64-QAM optical-fibre transmission data and define clustering, the stereographic projection and the necessary quantum concepts for the hybrid protocols. Next, Section 3 introduces the developed Stereographic Quantum *k*NN (SQ-*k*NN), while Section 4 defines the developed quantum-inspired 2D Stereographic Classical *k*NN (2DSC-*k*NN) algorithm and proves its equivalence to the SQ-*k*NN quantum algorithm. In Section 5, we describe the various experiments for testing the algorithms, present the obtained results, and discuss their conclusions. We conclude the main text in Section 6, proposing some directions for future research, some of which are further discussed in Appendix D.

### 1.1. Related Work

A unifying overview of several quantum algorithms is presented in [15] in a tutorial style. An overview targeting data scientists is provided in [16]. The idea of using quantum information processing methods to obtain speedups for the *k*-means algorithm was proposed in [17]. In general, neither the best nor even the fastest method for a given problem and problem size can be uniquely ascribed to either the class of quantum or classical algorithms, as observed in the detailed discussion presented in [9]. The advantages of using local (classical) processing units alongside quantum processing units in a distributed fashion are discussed in [18]. The accuracy of (quantum) *k*-means has been demonstrated experimentally in [19] and in [20], while quantum circuits for loading classical data into a quantum computer are described in [21].

An algorithm is proposed in [2] that solves the problem of clustering N-dimensional vectors to M clusters in O(log(MN)) time on a quantum computer, which is exponentially faster than the O(poly(MN)) time for the (then) best known classical algorithm. The approach detailed in [2] requires querying the QRAM [22] for preparing a ‘mean state’, which is then used to find the inner product between the centroid (by default, the mean point) using the SWAP test [23,24,25]. However, there exist some significant caveats to this approach. Firstly, this algorithm achieves an exponential speedup only when comparing the bit-to-bit processing time with the qubit-to-qubit processing time. If one compares the bit-to-bit execution times of both algorithms, the exponential speedup disappears, as shown in [4,26]. Secondly, since stable enough quantum memories do not exist, a hybrid quantum-classical approach must be used in real-world applications. Namely, all the information must be stored in classical memories, and the states to be used in the algorithm are prepared in real time. The process of preparing quantum states from classical data is known as ‘Data Embedding’ since we are embedding the classical data into quantum states. This, as mentioned before [4,26], slows down the algorithm to only a polynomial advantage over classical *k*-means. However, we propose an approach whereby this step of embedding can be treated as a data pre-processing step, allowing us to achieve some advantages in accuracy and convergence rate, and taking a step towards making the quantum approach more viable. Instead of using a quantum algorithm, classical alternatives mimicking their behaviour, collectively known as quantum-inspired algorithms, have shown much promise in classically achieving some types of advantage that are demonstrated by quantum algorithms [4,26,27,28], but as [9] remarks, the massive increase in runtime with rank, condition number, Frobenius norm, and error threshold make the algorithms proposed in [4,26] impractical for matrices arising from real-world applications. This observation is supported by [29].

Recent works such as [26] suggest that even the best QML algorithms, without state preparation assumptions, fail to achieve exponential speedups over their classical counterparts. In [4], it is pointed out that most QML algorithms are incomparable to classical algorithms since they take quantum states as input and output quantum states, and that there is no analogous classical model of computation where one could search for similar classical algorithms. In [4], the idea of matching state preparation assumptions with $\ell^{2}$-norm sampling assumptions (first proposed in [26]) is implemented by introducing a new input model, *sample and query access* (SQ access). In [4], the Quantum *k*-means algorithm described in [2] is ‘de-quantised’ using the ‘toolkit’ developed in [26], i.e., a classical quantum-inspired algorithm is provided that, with classical SQ access assumptions replacing quantum state preparation assumptions, matches the bounds and runtime of the corresponding quantum algorithm up to the polynomial slowdown. From the works [4,26,30], we can conclude that the exponential speedups of many quantum machine learning algorithms that are under consideration arise not from the ‘quantumness’ of the algorithms but instead from strong input assumptions, since the exponential part of the speedups vanish when classical algorithms are provided analogous assumptions. In other words, in a wide array of settings, these algorithms do not provide exponential speedups but rather yield polynomial speedups on classical data.

The fundamental aspect that allowed for the exponential speedup in [26] is exemplified by the problem of recommendation systems. The philosophy of classical recommendation algorithms before this breakthrough was to estimate all the possible preferences of a user and then suggest one or more of the most preferred objects. A quantum algorithm in [8] promised an exponential speedup but provided a recommendation without estimating all the preferences; namely, it only provided a *sample* of the most preferred objects. This process of sampling, along with state preparation assumptions, was, in fact, what gave the quantum algorithm an exponential advantage. The new classical algorithm also obtains comparable speedups by only providing samples rather than solving the whole preference problem. In [4], it is argued that the time taken to create the quantum state should be included for comparison since the time taken is not insignificant; it is also claimed that for every such linear algebraic quantum machine learning algorithm, a polynomially slower classical algorithm can be constructed by using the binary tree data structure described in [26]. Since then, more sampling algorithms have shown that multiple quantum exponential speedups are not due to the quantum algorithms themselves but due to the way data are provided to the algorithms and how the quantum algorithm provides the solutions [4,29,30,31]. Notably, in [31], it is argued that there exist competing classical algorithms for all linear algebraic subroutines and thus for many quantum machine learning algorithms. However, as pointed out in [9] and proven in [29], significant caveats exist to these aforementioned results of quantum-inspired algorithms. The polynomial factor in these algorithms often contains a very high power of the rank and condition number, making them suitable only for sparse low-rank matrices. Matrices of real-world data are often relatively high in rank and hence unfavourable for such sampling-based quantum-inspired approaches. Whether such sampling algorithms can be used also highly depends on the specific application and whether or not samples of the solution instead of the complete data are suitable. It should be pointed out that quantum machine learning algorithms generally do not provide an advantage if such complete data are needed.

The method of encoding classical data into quantum states contributes to the complexity and performance of the algorithm. An extensive analysis and testing of the hybrid quantum-classical implementation of the quantum *k*-means algorithm using angle embedding can be found in [10]. In this work, the use of the ISP is proposed. Others have explored this procedure [32,33,34] as well; however, the motivation, implementation, and use vary significantly, as well as the procedure for embedding data points into quantum states. There has also been no extensive testing of the proposed methods, especially not in an industry context. In our method, we exclusively use pure states from the Bloch sphere since this reduces the complexity of the application. Lemma 3 assures us that our method with existing quantum techniques is applicable for nearest neighbour clustering. In contrast, the density matrices of mixed states and the normalised trace distance between the density matrices are used for the binary classification in [32,33]. A crucial thing to consider here is to distinguish the contribution of the ISP from the quantum effects. We will see in Section 5 that the ISP seems to be the most important contributing factor. In [35], it is also proposed to encode classical information into quantum states using the ISP in the context of quantum generative adversarial networks. Their motivation for using the ISP is due to the fact that it is injective and can hence be used to uniquely represent every point in the 2D plane without any loss of information. On the other hand, angle embedding loses all amplitude information due to the normalisation of all points. A method to transform an unknown manifold into an n-sphere using ISP is proposed in [36]—here, however, the property of their concern was the conformality of the projection since subsequent learning is performed upon the surface. In [37], a parallelised version of [2] is developed using the FF-QRAM procedure [38] for amplitude encoding and the ISP to ensure a injective embedding.

In the method of Spherical Clustering [39], the nearest neighbour algorithm is explored based on the cosine similarity measure (Equation (21) and Lemma 2). The cosine similarity is used in cases of information retrieval, text mining, and data mining to find the similarity between document vectors. It is used in those cases because the cosine similarity has low complexity for sparse vectors since only the non-zero co-ordinates need to be considered. For our case as well, it is in our interest to study Equations (16)–(18) with the cosine dissimilarity. This approach becomes particularly relevant once we employ stereographic embedding to encode the data points into quantum states.

### 1.2. Contribution

In this work, we first develop generalised stereographic embedding for hybrid quantum-classical *k*NN clustering as a better encoding that allows the quantum algorithm (Section 3) to outperform the accuracy and convergence of classical *k*-means algorithm in the absence of noise; in contrast, angle embedding introduces fundamental limitations to the accuracy not due to quantum noise. To validate this statement, we simulate this algorithm classically, which translates into an equivalent classical quantum-analogous stereographic *k*NN clustering algorithm (Section 4). One must note that we do not demonstrate that running the stereographic quantum *k*NN algorithm is more practical than the classical *k*-means algorithm in the NISQ context. We show that stereographic quantum *k*NN clustering converges faster and is more accurate than other hybrid quantum-classical *k*NN algorithms with angle or amplitude embedding. In parallel, the benchmarking of the classical stereographic *k*NN algorithm lets us claim that for the problem of decoding 64-QAM optical-fibre signals, the generalised ISP and spherical centroid can allow for better accuracy and convergence.

The extensive testing upon the *real-world, experimental QAM dataset* (Section 2.1) revealed some significant results regarding the dependence of accuracy, runtime, and convergence performance upon the radius of projection, number of points, noise in the optical-fibre, and stopping criterion—described in Section 5. Noteworthy, we observe the existence of a finite optimal radius for the ISP (not equal to 1). To the best of our knowledge, no other work has considered a generalised projection radius for quantum embedding or studied its effect. Through our experimentation, we have verified that there exists an ideal radius greater than 1 for which accuracy performance is maximised. The advantageous implementation of the algorithm upon experimental data shows that our procedure is quite competitive. The fact that the developed quantum algorithm has an entirely classical analogue (with comparable time complexity to the classical *k*-means algorithm) is a distinct advantage in terms of in-field deployment, especially compared to [2,9,17,32,33,34,37]. The developed quantum algorithm also has another advantage in the context of Noisy Intermediate-Scale Quantum (NISQ) realisations—it has the least circuit depth and circuit width among all candidates [2,9,34,37]—making it easier to implement with the current quantum technologies. Another significant contribution is our generalisation of the dissimilarity for clustering; instead of Euclidean dissimilarity (distance), we consider other dissimilarities which might be better estimated by quantum circuits (Appendix E). A somewhat similar approach was developed in parallel by [40] in the context of amplitude embedding. All previous approaches [2,9,34,37] only try to estimate the Euclidean distance. We also make the contribution of studying the relative effect of ‘quantumness’ and the ISP, something completely overlooked in previous works. We show that the quantum ‘advantage’ in accuracy performance touted by works such as [32,33,34,37] is in reality quite suspect and achievable through classical means. In Appendix D, we describe a generalisation of the stereographic embedding—the Ellipsoidal embedding, which we expect to provide even better results in future works.

Other secondary contributions of our work include:The development of a mathematical formalism for the generalisation of *k*NN to indicate the contribution of various parameters such as dissimilarities and dataspace (Section 2.4);Presenting the procedure and circuit for *stereographic embedding* using the Bloch embedding procedure, which consumes only *O(1) in time and resources* (Section 3.1).

## 2. Preliminaries

In this section, for completeness, we touch upon some concepts and background information required to understand this paper. These concepts range from small general statements on quantum states (Bloch sphere, fidelity, and Bell-state measurement in Section 2.2 and Section 2.3) to the mathematical formalism of *k*NN (Section 2.4, Section 2.5 and Section 2.6), and stereographic projection (Section 2.7). We begin by first describing the optic-fibre experimental setup used to collect the 64-QAM dataset, upon which the clustering algorithms were tested and benchmarked.

### 2.1. Optical-Fibre Setup

M-ary Quadrature Amplitude Modulation (M-QAM) is a simple and popular protocol for digital data transmission through analog communication channels. It is widely used in optical-fibre communication networks, and the decoding process of the received data often uses the *k*-nearest-neighbour algorithm to cluster nearby points. More details, including the description of the model used in the experiments, can be found in Appendix A. We now describe the experimental setup used to collect the dataset that is used for benchmarking the clustering algorithms.

The dataset contains a launch-power (laser-power feed into the fibre) sweep (four datasets collected at four different launch powers) of 80 km fibre transmission of coherent dual polarization (DP)-64QAM with a gross data rate of $960\times 10^{9}$ bits/s. For the dataset, we assumed 15% overhead for forward error correction (FEC) and used 3.47% overhead for pilots and training sequences; thus, the net bit rate is $800\times 10^{9}$ bits/s. Note that the pilots and training sequences are removed after the MIMO equalizer. An overview of the experimental setup [10,14] to capture this real-world database is shown in Figure 1. Four $120\times 10^{9}$ Samples/s digital-to-analog converters (DACs) generate an electrical signal amplified by four 60 GHz 3dB-Bandwidth amplifiers. A tunable 100 kHz external cavity laser (ECL) source generates a continuous wave signal that is modulated by a 32 GHz DP-I/Q modulator. The receiver comprises an optical 90${}^{\circ }$-hybrid and four 100 GHz balanced photodiodes. The electrical signals are digitized using four 10-bit analog-to-digital converters (ADCs) with $256\times 10^{9}$ Samples/s and 110 GHz. Subsequently, the raw signals are pre-processed by the receiver digital signal processing (DSP) blocks.

The datasets were collected in a very short time, corresponding to the memory size of the oscilloscope, which is limited. This is referred to as offline processing. At the receiver, the signals were normalised to fit the alphabet. The average launch power in watts can be calculated as follows:$$\begin{array}{c} P_{\left(W\right)}=1W\cdot 10^{\left(P_{\left(dBm\right)})/10\right)}/1000=10^{(P_{\left(dBm\right)}-30)/10} \end{array}$$There are four sets of published data with different launch powers, corresponding to different levels of non-linear distortions during transmission: 2.7 dBm, 6.6 dBm, 8.6 dBm, and 10.7 dBm. Each dataset consists of the ‘alphabet’ (initial analog transmission values), the error-corrected received analog values, and the true labels of the transmitted points. The data have been explained and visualised in detail in the Appendix A. To quantify the system performance of an amplified coherent optical communication system, one uses either a launch power sweep or an OSNR (Optical Signal to Noise Ratio) sweep. While the OSNR metric is used when the system is operating in the linear region, the launch power is the preferred metric to show the performance degradation in the nonlinear region since the induced nonlinear effects are directly proportional to the launch power. The signal-to-noise ratio (SNR) for each launch power can be computed using the following expression, where z are the received noisy signals and x are the noiseless target symbols (the launched signals):$$\begin{array}{cc} \mathrm{SNR} & =10\log_{10}\left(\frac{\mathrm{mean}({∥z∥}^{2})}{\mathrm{mean}({∥z-x∥}^{2})}\right). \end{array}$$

After obtaining this noisy real-world dataset, our task is to decode the received analog values into bit-strings. The *k*NN is the candidate of choice since it classifies datasets into clusters by associating an ‘average point’ (centroid) to each cluster. In our method, the objective of the clustering algorithm is first to identify, using the set of received signals, a given number *M* of centroids (one for each cluster) and then to assign each signal to the ‘nearest’ centroid. The second step is classification. This creates the clusters, which can then be decoded into bit signals through the process of demapping. Demapping consists of mapping the original transmission constellation (alphabet) to the current centroids and then assigning the bit-string label associated with that initial transmission point to all the points in the cluster of that centroid. This process completes the final step of the QAM protocol, translating the analog values to bit-strings read by the receiver. The size *M* of the constellation is known since we know beforehand which QAM protocol is being used. We also know the “alphabet”, i.e., the initial and ideal points at which the signals were transmitted.

### 2.2. Bloch Sphere

It is well known that all the qubit pure states can be obtained from the zero state using the unitary *U* [41]
(1)$$\begin{array}{cc} |\psi (\theta ,\phi )\rangle & =U\left(\theta ,\phi \right)|0\rangle =cos\left(\theta /2\right)|0\rangle +e^{i\phi }\sin\left(\theta /2\right)|1\rangle \end{array}$$
where
(2)$$\begin{array}{c} U\left(\theta ,\phi \right):=\left(\begin{array}{cc} \cos\frac{\theta }{2} & -\sin\frac{\theta }{2}\\ e^{i\phi }\sin\frac{\theta }{2} & e^{i\phi }\cos\frac{\theta }{2} \end{array}\right)\;. \end{array}$$These are unit vectors in the unit sphere of $C^{2}$, but it is also well known that the corresponding density matrices are uniquely represented by the Bloch vectors
a(θ,ϕ):=(sinθcosϕ,sinθsinϕ,cosθ)
as points in the unit sphere $S^{2}\left(1\right)\subset R^{3}$ [41] (the Bloch sphere) through the relation
(3)$$\begin{array}{cc} \rho (\theta ,\phi ) & =\left|\psi (\theta ,\phi )\rangle \langle \psi (\theta ,\phi )\right|\\ & =\left(\begin{array}{cc} \cos^{2}\frac{\theta }{2} & -e^{-i\phi }\cos\frac{\theta }{2}\sin\frac{\theta }{2}\\ e^{i\phi }\cos\frac{\theta }{2}\sin\frac{\theta }{2} & \sin^{2}\frac{\theta }{2} \end{array}\right)\\ & =\frac{1}{2}\left(\begin{array}{cc} 1+\cos\theta & -e^{-i\phi }\sin\theta \\ e^{i\phi }\sin\theta & 1-\cos\theta \end{array}\right)\\ & =\frac{1}{2}\left(1+a\left(\theta ,\phi \right)\cdot \vec{\sigma }\right) \end{array}$$
where 1 is the identity matrix and $\vec{\sigma }=\left(\sigma_{x},\sigma_{y},\sigma_{z}\right)$ is the vector of Pauli matrices
$$\begin{array}{cccccc} \sigma_{x} & =\left(\begin{array}{cc} 0 & 1\\ 1 & 0 \end{array}\right)\; & \sigma_{y} & =i\left(\begin{array}{cc} 0 & -1\\ 1 & 0 \end{array}\right)\; & \sigma_{x} & =\left(\begin{array}{cc} 1 & 0\\ 0 & -1 \end{array}\right). \end{array}$$

Regarding mixed states, notice that Equation (3) is linear and thus, convex combinations of density matrices translate to convex combinations of Bloch vectors, meaning that the interior of the sphere represents the mixed states. Namely, the most general qubit quantum states can be represented by
(4)$$\begin{array}{cccc} \rho_{a}\equiv \rho \left(a\right) & =\frac{1}{2}\left(1+a\cdot \vec{\sigma }\right),\; & \; & {∥a∥}_{2}\leq 1. \end{array}$$Finally, since the Pauli matrices are orthogonal operators under the Hilbert-Schmidt inner product, this inner product is easily computed as
(5)$$\begin{array}{cc} \mathrm{Tr}\left(\rho_{a_{1}}\rho_{a_{2}}\right) & =\frac{1}{2}\left(1+a_{1}\cdot a_{2}\right). \end{array}$$
which for pure states coincides with the fidelity.

Using the Bloch sphere representation of qubit quantum states also makes it easy to find orthogonal states and compute diagonalizations. Indeed, let a be a unit vector ($∥a∥=a\cdot a=1$), thus representing the pure state $\rho_{a}=\frac{1}{2}\left(1+a\cdot \vec{\sigma }\right)$, then the orthogonal state to $\rho_{a}$ is simply the antipodal point
(6)$$\begin{array}{c} \rho_{-a}=\frac{1}{2}\left(1-a\cdot \vec{\sigma }\right) \end{array}$$
which can be shown by computing the inner product as in Equation (5)
(7)$$\begin{array}{c} \mathrm{Tr}\left(\rho_{+a}\rho_{-a}\right)=\frac{1}{4}\left(1+a\cdot (-a)\right)=0. \end{array}$$Hence, *the Bloch eigenvectors* for any Bloch vector a are $\pm \frac{a}{∥a∥}$, the two antipodal points where the line of a intersect the Bloch sphere. Namely, for any mixed quantum state corresponding to the Bloch vector a, we can decompose the quantum state as
(8)$$\begin{array}{cc} \frac{1}{2}\left(1+a\cdot \vec{\sigma }\right) & =p\frac{1}{2}\left(1+\frac{a}{∥a∥}\cdot \vec{\sigma }\right)+\left(1-p\right)\frac{1}{2}\left(1-\frac{a}{∥a∥}\cdot \vec{\sigma }\right)\\ & =\frac{1}{2}\left(1+\left(2p-1\right)\frac{a}{∥a∥}\cdot \vec{\sigma }\right) \end{array}$$
with
(9)$$\begin{array}{ccc} 2p-1 & =∥a∥\; & \Rightarrow p=\frac{1}{2}\left(1+∥a∥\right). \end{array}$$

In the next section, we discuss how we use the Bell-state measurement to estimate the fidelity between quantum states and exposit when this should be chosen over the SWAP test.

### 2.3. Bell-State Measurement and Fidelity

We use the Bell-state measurement to estimate the fidelity between two pure states. The Bell-state measurement is defined as the von-Neumann measurement of the maximally entangled basis
(10)$$|\phi_{ij}\rangle :=CNOT(H⊗1)|ij\rangle ,$$
which by construction is equivalent to a standard basis measurement after (H⊗1)CNOT as displayed in Figure 2. This measurement can be used to estimate the fidelity as follows.

**Lemma** **1.**
*Let |ψ〉 and |χ〉 be two qubit pure states and let $|\phi_{11}\rangle :=CNOT(H⊗1)|11\rangle$ (the singlet Bell state). Then*

(11)
$$\begin{array}{cc} {\left|\langle \phi_{11}|\left(|\psi \rangle ⊗|\chi \rangle \right)\right|}^{2} & =\frac{1}{2}{\left(1-|\langle \psi |\chi \rangle \right|}^{2}). \end{array}$$



**Proof.** Let us write the states as
(12)$$\begin{array}{cccc} |\psi \rangle & =\psi_{0}|0\rangle +\psi_{1}|1\rangle \; & |\chi \rangle & =\chi_{0}|0\rangle +\chi_{1}|1\rangle \;, \end{array}$$Then, the state before the standard-basis measurement is
(13)$$\begin{array}{cc} |\psi_{\mathrm{out}}\rangle =\left(H⊗1\right)CNOT\left(|\psi \rangle ⊗|\chi \rangle \right)\; & =\frac{1}{\sqrt{2}}\left(\begin{array}{c} \psi_{0}\chi_{0}+\psi_{1}\chi_{1}\\ \psi_{0}\chi_{1}+\psi_{1}\chi_{0}\\ \psi_{0}\chi_{0}-\psi_{1}\chi_{1}\\ \psi_{0}\chi_{1}-\psi_{1}\chi_{0} \end{array}\right) \end{array}$$
and in particular, the probability of outcome ij=11 (i.e., simultaneous measurement of both qubits yields value ‘1’ on each qubit) can be written as
(14)$$\begin{array}{cc} {\left|\langle \phi_{11}|\left(|\psi \rangle ⊗|\chi \rangle \right)\right|}^{2} & ={\left|\langle 11|\psi_{out}\rangle \right|}^{2}=\frac{1}{2}{\left|\psi_{0}\chi_{1}-\psi_{1}\chi_{0}\right|}^{2}. \end{array}$$The fidelity is obtained now by adding and subtracting $\psi_{0}^{*}\psi_{0}\chi_{0}^{*}\chi_{0}+\psi_{1}^{*}\psi_{1}\chi_{1}^{*}\chi_{1}$ and computing
(15)$$\begin{array}{cc} {\left|\langle \phi_{11}|\left(|\psi \rangle ⊗|\chi \rangle \right)\right|}^{2} & =\frac{1}{2}\left(1-\psi_{1}\chi_{0}\psi_{0}^{*}\chi_{1}^{*}-\psi_{0}\chi_{1}\psi_{1}^{*}\chi_{0}^{*}-\psi_{0}^{*}\psi_{0}\chi_{0}^{*}\chi_{0}-\psi_{1}^{*}\psi_{1}\chi_{1}^{*}\chi_{1}\right)\\ & =\frac{1}{2}\left(1-\right|\psi_{0}^{*}\chi_{0}+\psi_{1}^{*}\chi_{1}{|}^{2})\\ & =\frac{1}{2}{\left(1-|\langle \psi |\chi \rangle \right|}^{2}), \end{array}$$
concluding the proof.    □

Lemma 1 is used to construct the quantum clustering algorithm in Section 3. We will use the quantum circuit of Figure 2 for the fidelity estimation in the developed quantum algorithm.

**Remark** **1.**
*Since we are only interested in the ij=11 outcome and we are measuring qubits, the course-grained projective measurement defined by $\left|\phi_{11}\rangle \langle \phi_{11}\right|$ and $1-\left|\phi_{11}\rangle \langle \phi_{11}\right|$ is sufficient for computing the inner product. The non-destructive version of this measurement is known as the **SWAP test** [24,25], first described in [23]. This test has been used extensively for overlap estimation in quantum algorithms [2]. The SWAP test requires one to only measure an ancilla qubit instead of the two input qubits, leaving them in the post-measurement state, which can be used later for other purposes. However, given the current limitations of NISQ technologies, storing quantum information for reuse is quite impractical; therefore, we prefer the destructive measurement version for overlap estimation. Namely, we use the Bell-state measurement instead of the SWAP test because the post-measurement state is unnecessary.*


### 2.4. Nearest-Neighbour Clustering Algorithms

Clustering is a simple, powerful and well-known machine-learning algorithm that has been extensively used throughout the literature. In this section, we summarise some standard and basic notions introduced by clustering and define this class of heuristic algorithms precisely so that we can make clear the difference between regular clustering and the quantum and quantum-inspired clustering algorithms introduced in this paper. We first define the involved variables needed for the *k*NN.

**Definition** **1**(Clustering State)**.** *We define a k**-Nearest-Neighbour Clustering State**, or **clustering state** for short, as a collection $\left(D,\bar{c},D,d\right)$ where*
*D is a space called **dataspace** with elements called **points**.**D⊆D is a subset called **dataset** consisting of points called **datapoints***.*$\bar{c}=\left(c_{1}\;c_{2}\;\ldots \;c_{k}\right)\subseteq D^{k}$ is a list (of size k) of points called **centroids****d:D×D⟼R is a lower bounded function called **dissimilarity function**, or **dissimilarity** for short.*

Note that *d* does not have to be a distance metric. We now define the basic steps that are repeated in the clustering algorithm.

**Definition** **2**(Clusters and Centroid update)**.** *Let $\left(D,\bar{c},D,d\right)$ be a clustering state. We define the **clusters** of the state as, for each j=1,…,k, the set*
(16)$$\begin{array}{c} C_{j}\left(\bar{c}\right)=\left\{p\in D\;|\;d\left(p,c_{j}\right)\leq d\left(p,c_{\ell }\right)\;\forall \;\ell =1,\ldots ,k,\;p\notin \underset{\ell <j}{⋃}C_{\ell }\left(\bar{c}\right)\right\}. \end{array}$$*We now define the **possible new centroids** of a subset C⊆D as the set*
(17)$$\begin{array}{cc} P(C):= & \underset{x\in D}{\mathrm{argmin}}\sum_{p\in C}d\left(x,p\right) \end{array}$$*of all points minimising the total (and thus the average) dissimilarity. Then, we call a **centroid update** any function $c^{\mathrm{update}}:P\left(D\right)\to D$ (where P denotes the power set) of clusters such that $c^{\mathrm{update}}\left(C\right)$ is a possible new centroid, namely such that $c^{\mathrm{update}}\left(C\right)\in P\left(C\right)$, for all j=1,…,k. We then define the following short-hand notation for the centroid update of $\bar{c}$, namely the new list of centroids*
(18)$$\begin{array}{cc} {\bar{c}}^{\mathrm{update}}\left(\bar{c}\right)\; & =\left(c^{\mathrm{update}}\left(C_{1}\left(\bar{c}\right)\right),\ldots ,c^{\mathrm{update}}\left(C_{k}\left(\bar{c}\right)\right)\right). \end{array}$$

We now define the general *k*-nearest-neighbour clustering algorithm.

**Definition** **3**(*K*-Nearest-Neighbour Clustering Algorithm (*k*NN))**.** *Finally, we define a K**-Nearest-Neighbour clustering algorithm (kNN)** as a pair of clustering state and centroid update $\left(D,{\bar{c}}_{1},D,d,{\bar{c}}^{\mathrm{update}}\right)$. The kNN algorithm defines a sequence of clustering states $\left(D,{\bar{c}}_{i},D,d\right)$ via ${\bar{c}}_{i+1}={\bar{c}}^{\mathrm{update}}\left({\bar{c}}_{i}\right)$ for all i∈N which we call the **iterations** (of the algorithm).*

A point of note is that Equation (17) implies that the new centroid is one of the points x in the dataspace that minimises the total (and hence the average) dissimilarity with all the points p in the cluster. Moreover, notice that this definition requires one to initialise or populate the list $\bar{c}$ with initial values, i.e., the *initial centroids*
${\bar{c}}_{1}$ must be defined as a starting point for the clustering algorithm. The initial centroids can be assigned randomly or defined as a function of parameters such as the dataset.

Another comment about Equation (17): in our case, we will see later in Section 3.2 that all choices of points from the set $P_{j}$ will be equivalent. As in our algorithm, this freedom of choice can be exploited to reduce the amount of computation or for other optimisations.

Notice that Equations (17) and (18) implies that centroids are generally not part of the original dataset; however, according to Equations (17) and (18), they must be restricted to the space in which the dataset is defined. Definitions involving centroids for which $\bar{c}\notin D^{k}$ are possible but are not used in this work.

One can observe that any *k*NN can be broken down into two steps that keep alternating until a stopping condition (a condition which, when true, forces the algorithm to terminate) is met: a *cluster update* which updates the points associated with the newly calculated centroid, and then a *centroid update* which recalculates the centroid based upon the new points associated to it through its cluster. For the cluster update, the value of the centroid calculated in the previous iteration is taken, and its cluster set is constructed by collecting all the points in the dataset that are ‘closer’ to it than any other centroid. The ‘closeness’ is computed by using a pre-defined dissimilarity. In the next step, the centroids are updated by searching in the dataspace, and for each updated cluster, a new point for which the sum of dissimilarities between that point and all points in the cluster is minimised.

This procedure will lead to different results if one changes the dissimilarity or the space of data points or both. In this paper, we explore the effects of changing this dissimilarity as well as the space of data points, and we shall explain it in the context of quantum states.

### 2.5. Euclidean Dissimilarity and Classical Clustering

It can be observed from the centroid update in Equations (16) and (17) that the dissimilarity plays a central role in the clustering algorithm. The nature of this function directly controls the first step of the cluster update since the dissimilarity is used to compute the ‘closeness’ between any two points in the dataspace. It is also apparent that if the dissimilarity is changed in the centroid update, the points at which the minimum is achieved could also change.

The Euclidean dissimilarity $d_{e}:R^{n}\times R^{n}\to R$ is defined simply as the square of the Euclidean distance between the points:(19)$$\begin{array}{c} d_{e}\left(a,b\right)={∥a-b∥}^{2}\;. \end{array}$$For a finite subset $C\subset R^{n}$, the minimisation of Equations (17) and (18) yields a unique point, reducing to the average point of the cluster: (20)$$\begin{array}{c} c_{e}^{\mathrm{update}}\left(C\right):=\underset{x\in R^{n}}{\mathrm{argmin}}\sum_{p\in C}d_{e}\left(x,p\right)=\frac{1}{\left|C\right|}\sum_{p\in C}p\;, \end{array}$$
which we call the *Euclidean centroid update*. This is the most typical case of the centroid update, where the new centroid is updated as the mean point of all points in the cluster. This corresponds to the classic *k*-means clustering algorithm [42], which now can be defined as follows.

**Definition** **4**(*n*-Dimensional Euclidean Classical *k*NN (*n*DEC-*k*NN))**.** *An n**-dimensional classical Euclidean kNN algorithm** is any clustering algorithm with dataspace $R^{n}$, Euclidean dissimilarity and cluster update as the average point, as in Equation *(20)*. Namely, any clustering algorithm of the form $\left(D,\bar{c},R^{n},d_{e},c_{e}^{\mathrm{update}}\right)$.*

The computation of the centroid through Equation (20) instead of Equations (17) and (18) reduces the complexity of the centroid update step; such a reduced expression is used to compute the updated centroids rather than searching the entire dataspace for the minimising points during the centroid update.

### 2.6. Cosine Dissimilarity

In this work, we project the collected two-dimensional dataset (described in Section 2.1) into a sphere via the ISP. After this projection, the calculation of the centroids according to Equation (20) would generally yield centroids which lie inside the sphere instead of on the $S^{2}\left(r\right)$ surface due to the convex nature of the sphere’s surface.

In our work, to use qubit pure states, we restrict the dataspace D to the sphere surface $S^{2}\left(r\right)$, forcing the centroids to lie on the surface of a sphere. This naturally leads to the question of what the proper reformulation of Equations (17) and (18) is, and whether a computationally inexpensive formula similar to Equation (20) exists for this case as well. This question will be answered in Lemma 3. For this purpose, it is useful to first define the cosine dissimilarity [43] and see how it relates to the Euclidean dissimilarity.

**Definition** **5**(Cosine Dissimilarity)**.** *For two points, a and b in an inner-product space D, the **cosine dissimilarity**, is defined as:*
(21)$$\begin{array}{c} d_{s}\left(a,b\right)=1-\frac{a\cdot b}{∥a∥∥b∥}, \end{array}$$*where a·b is the inner product between the two points expressed as vectors from the origin, $∥a∥$ is the norm of a induced by the inner product.*

This is called *cosine* dissimilarity because when $a,b\in R^{n}$ the cosine dissimilarity $d_{s}\left(a,b\right)$ reduces to 1−cos(α), where α is the angle between a and b. The cosine dissimilarity is also known sometimes as *cosine distance* (although it is not a distance), while $\frac{a\cdot b}{∥a∥∥b∥}$ is well known as *cosine similarity*. This quantity, by construction, only depends on the direction of the vectors and not their magnitude. Said otherwise, we have
(22)$$\begin{array}{c} d_{s}\left(a,b\right)=d_{s}\left(ca,b\right)=d_{s}\left(a,cb\right) \end{array}$$
for any positive constant c>0. We also note that the cosine dissimilarity of Equation (21) can be related to the Euclidean dissimilarity of Equation (19) if a and b lie on the n-sphere $S^{n}(r):=\left\{\;s\in R^{n+1}\;|\;{∥s∥}_{2}=r\;\right\}$ of radius *r*, as stated by the following lemma.

**Lemma** **2.**
*Let $d_{s}$ and $d_{e}$ be the cosine and Euclidean dissimilarities, respectively. Let $s_{1}$, $s_{2}$
$\in S^{n}\left(r\right)$ be points on the n-sphere of radius r, then*

(23)
$$\begin{array}{c} d_{e}\left(s_{1},s_{2}\right)=2r^{2}d_{s}\left(s_{1},s_{2}\right)\;. \end{array}$$



**Proof.** Assuming $s_{1}$, $s_{2}$$\in S^{n}\left(r\right)$, Equation (21) reduces to:
(24)$$\begin{array}{cc} d_{s}\left(s_{1},s_{2}\right) & =1-\frac{1}{r^{2}}s_{1}\cdot s_{2}, \end{array}$$
then
(25)$$\begin{array}{cc} 2r^{2}d_{s}\left(s_{1},s_{2}\right) & =2r^{2}-2s_{1}\cdot s_{2}={∥s_{1}∥}^{2}+{∥s_{2}∥}^{2}-2s_{1}\cdot s_{2}={∥s_{1}-s_{2}∥}^{2}=d_{e}\left(s_{1},s_{2}\right), \end{array}$$
concluding the proof.    □

From this, we can expect that the minimiser of the centroid update equation (Equation (17)) computed using the cosine dissimilarity will closely relate to the Euclidean centroid update. However, the derivation is not straightforward since the Euclidean centroid update does not lie on the same sphere, but lies inside at a smaller radial distance. This is shown in the following lemma.

**Lemma** **3.**
*Let $C\subset S^{n}\left(r\right)$ be a finite set, then*

(26)
$$\begin{array}{cc} c_{s}^{\mathrm{update}}\left(C\right):= & \underset{x\in S^{n}\left(r\right)}{\mathrm{argmin}}\sum_{p\in C}d_{s}\left(x,p\right)=r\frac{\sum_{p\in C}p}{∥\sum_{p\in C}p∥}. \end{array}$$

*We call this the **cosine** or **spherical centroid update**. In particular, thus*

(27)
$$\begin{array}{cc} c_{s}^{\mathrm{update}}\left(C\right) & =r\frac{c_{e}^{\mathrm{update}}\left(C\right)}{∥c_{e}^{\mathrm{update}}\left(C\right)∥} \end{array}$$

*where $c_{e}^{\mathrm{update}}\left(C\right)=\frac{1}{\left|C\right|}\sum_{p\in C}p$ is the Euclidean centroid update of Equation *(20)*.*


**Proof.** The second claim is trivial; we thus have to prove only the first claim. Given that $C\subseteq S^{n}\left(r\right)$, then, according to Lemma 2, the cosine dissimilarity given in Equation (21) reduces for all a,b∈C to:
(28)$$\begin{array}{cc} d_{s}\left(a,b\right) & =1-\frac{a\cdot b}{∥a∥∥b∥}=1-\frac{1}{r^{2}}a\cdot b. \end{array}$$The minimisation in Equation (17) can then be calculated for the cosine dissimilarity with a Lagrangian (see Equation (31)) that satisfies Equations (17) and (18) at the minimising point. Namely, we have to find $x\in R^{n}$ that minimises
(29)$$\begin{array}{c} f\left(x\right)=\sum_{p\in C}d\left(x,p\right), \end{array}$$
subject to the restriction condition that assures that $x\in S^{n}\left(r\right)$, that is
(30)$$\begin{array}{c} g\left(x\right)={∥x∥}^{2}-r^{2}=0\;. \end{array}$$Such a Lagrangian is expressed as
(31)$$\begin{array}{cc} L(x,\lambda ) & =f\left(x\right)-\lambda g\left(x\right)=\sum_{p\in C}d_{s}\left(x,p\right)-\lambda \left({∥x∥}^{2}-r^{2}\right), \end{array}$$
where λ is the Lagrangian multiplier. We then calculate the centroid update by employing the derivative criteria to Equation (31).
(32)$$\begin{array}{cc} 0 & =\nabla \left(\sum_{p\in C}\left(1-\frac{1}{r^{2}}x\cdot p\right)-\lambda {∥x∥}^{2}+r^{2}\right)=-\frac{1}{r^{2}}\sum_{p\in C}p-2\lambda x \end{array}$$Therefore, the following holds:
(33)$$\begin{array}{cc} x & =-\frac{1}{2\lambda r^{2}}\sum_{p\in C}p\;. \end{array}$$Substituting Equation (33) into the restriction in Equation (30), we obtain the multiplier λ as:
(34)$$\begin{array}{cc} \left|\lambda \right| & =\frac{1}{2r^{3}}\cdot ∥\sum_{p\in C}p∥\;. \end{array}$$Therefore, the critical point and minimising point $c_{s}$ is written as
(35)$$\begin{array}{c} c_{s}=\frac{r}{∥\sum_{p\in C}p∥}\sum_{p\in C}p\;, \end{array}$$
as claimed.    □

We can observe that Lemma 3 implies that the minimiser obtained by restricting the point to lie on the surface of the sphere is the projection (from the origin) of the minimiser of the Euclidean dissimilarity into the sphere’s surface.

**Corollary** **1.**
*Let $C\subset S^{n}\left(r\right)$ be a finite set, and the possible new centroids of C under cosine dissimilarity in $R^{3}$ are*

(36)
$$\begin{array}{cc} P_{s}\left(C\right):= & \underset{x\in R^{n}}{\mathrm{argmin}}\sum_{p\in C}d_{s}\left(x,p\right)=\left\{r\sum_{p\in C}p:r>0\right\}. \end{array}$$

*We call these the **cosine possible new centroids**.*


**Proof.** We have
(37)$$\begin{array}{cc} c_{s}^{\mathrm{update}}:= & \underset{x\in R^{n}\setminus \left\{0\right\}}{\mathrm{argmin}}\sum_{p\in C}d_{s}\left(x,p\right)=\underset{x\in S^{n}\left(r\right),r>0}{\mathrm{argmin}}\sum_{p\in C}d_{s}\left(x,p\right)\\ & =\left\{r\frac{\sum_{p\in C}p}{∥\sum_{p\in C}p∥}:r>0\right\}=\left\{r\sum_{p\in C}p:r>0\right\}, \end{array}$$
where the last equality follows from Equation (22), namely
(38)$$\begin{array}{c} d_{s}\left(r\frac{\sum_{p\in C}p}{∥\sum_{p\in C}p∥},p\right)=d_{s}\left(\sum_{p\in C}p,p\right) \end{array}$$
for all *r* and p; thus making all the points $r\sum_{p\in C}p$, r>0 equivalent possibilities for the centroid update.    □

### 2.7. Stereographic Projection

The inverse stereographic projection (ISP), shown in Figure 3, is a bijective mapping
(39)$$\begin{array}{c} s_{r}^{-1}:R^{n}\mapsto S^{n}\left(r\right)\setminus \left\{N\right\} \end{array}$$
from the Euclidean space $R^{n}$ into an *n*-sphere $S^{n}\left(r\right)\subset R^{n+1}$ without the north pole *N*.

This mapping is interesting because of the natural equivalence between the 3D unit sphere $S^{2}\left(1\right)$ and the Bloch sphere of qubit quantum states. In this case, as displayed in Figure 3, the ISP maps a two-dimensional point $p=\left(p_{x},p_{y}\right)\in R^{2}$ into a three-dimensional point $s_{r}^{-1}\left(p\right)=\left(s_{x}\left(p\right),s_{y}\left(p\right),s_{z}\left(p\right)\right)\in S^{2}\left(r\right)\setminus \left\{\left(0,0,r\right)\right\}$ through the following set of transformations:(40)$$\begin{array}{cccccc} s_{x}\left(p\right) & =p_{x}\cdot \frac{2r^{2}}{p_{x}^{2}+p_{y}^{2}+r^{2}}\; & s_{y}\left(p\right) & =p_{y}\cdot \frac{2r^{2}}{p_{x}^{2}+p_{y}^{2}+r^{2}}\; & s_{z}\left(p\right) & =r\cdot \frac{p_{x}^{2}+p_{y}^{2}-r^{2}}{p_{x}^{2}+p_{y}^{2}+r^{2}}\\ \; & =p_{x}\cdot \frac{2r^{2}}{{∥p∥}^{2}+r^{2}}\; & \; & =p_{y}\cdot \frac{2r^{2}}{{∥p∥}^{2}+r^{2}}\; & \; & =r\cdot \frac{{∥p∥}^{2}-r^{2}}{{∥p∥}^{2}+r^{2}}\;. \end{array}$$The polar and azimuthal angles of the projected point are given by the expressions:(41)$$\begin{array}{cccc} \phi (p) & =\tan^{-1}\left(\frac{p_{y}}{p_{x}}\right)\; & \theta (p) & =2\cdot \tan^{-1}\left(\frac{r}{∥p∥}\right) \end{array}$$

This information, particularly Equation (41), will allow us to associate each point in $R^{2}$ to a unique quantum state through the Bloch sphere. Still, the inverse stereographic projection does not need to be bound to the preparation of quantum states and can be used as a transformation between classical *k*NN algorithms. Indeed, we can stereographically project and then perform classical clustering on the 3D data, and namely perform 3DEC-*k*NN as defined in Definition 4.

**Definition** **6**(Three-Dimensional Stereographic Classical *k*NN (3DSC-*k*NN))**.** *Let $s_{r}^{-1}$ be an ISP, and let $\left(D,\bar{c},R^{2},d_{e}\right)$ be a clustering state (recall, D is the dataset and $\bar{c}$ are the initial centroids). We then define the **3D Stereographic Classical kNN (3DSC-kNN)** as ($s_{r}^{-1}\left(D\right)$, $s_{r}^{-1}\left(\bar{c}\right)$, $R^{3}$, $d_{e},c_{e}^{\mathrm{update}})$.*

Here, we apply $s_{r}^{-1}$ elementwise and thus $s_{r}^{-1}\left(\bar{c}\right)=\left(s_{r}^{-1}\left(c_{1}\right),\ldots ,s_{r}^{-1}\left(c_{k}\right)\right)$ for any list of centroids $\bar{c}$ and $s_{r}^{-1}\left(D\right)=\left\{\;s_{r}^{-1}(p):p\in C\;\right\}$ for any set of points *C*, and where $C_{j}$ are the clusters as defined in Equation (16).

**Remark** **2.**
*Derivations and further observations can be found in Appendix C. Of particular note, as explained in more detail in Section C.2, is that changing the plane’s distance from the centre of the sphere is equivalent to a change of radius. Therefore, we can limit our analysis to projections where the centre of the plane is also the centre of the sphere without a loss of generality.*


## 3. Stereographic Quantum Nearest-Neighbour Clustering (SQ-kNN)

This section proposes and describes the quantum *k*NN algorithm using stereographic embedding. In Section 4, we demonstrate an equivalent quantum-inspired (classical) version. Section 3.1 defines the method to convert the classical data into quantum states. In what follows, we describe how these states are manipulated so that we obtain an output that can be used to perform clustering, using the circuit of Section 2.3 for the dissimilarity estimation. Section 3.2 defines the quantum algorithm in terms of Definitions 1–3, and Section 3.3 discusses the complexity and scalability of the algorithm. Section 3.4 discusses the SQ-*k*NN algorithm in the context of mixed states.

### 3.1. Stereographic Embedding, Bloch Embedding and Quantum Dissimilarity

For quantum algorithms to work on classical data, the classical data must be converted into quantum states. This process of encoding classical data into quantum states is also called *embedding*. The embedding of classical data into quantum states is not unique, and each technique’s pros and cons must be weighed in the context of a specific application. The process of data embedding is an active field of research. More details on existing embedding can be found in Appendix B.

Here, we propose the *stereographic embedding* as an improved embedding of classical vector $p\in R^{2}$ into a quantum state using its stereographic projection. We can split stereographic embedding into two steps: inverse stereographic projection and Bloch embedding. We define Bloch embedding, a variation of angle embedding, as follows.

**Definition** **7**(Bloch embedding)**.** *Let $P\in R^{3}$. We define the **Bloch embedded quantum state**, or **Bloch embedding** for short, of P as the quantum state*
(42)$$\psi_{P}:=\frac{1}{2}\left(1+\frac{P}{∥P∥}\cdot \vec{\sigma }\right)$$*which is simply the pure state obtained using $P/∥P∥$ as the Bloch vector.*

At this point, we define this general embedding for general three-dimensional points since this general form will yield the quantum dissimilarity defined next. We will also define this embedding in the context of the ISP in Definition 9, below.

To obtain $\psi_{P}$, the state can be encoded as explained in the preliminaries in Section 2.2, through Equations (2) and (3). For Bloch embedding, the θ and ϕ of Equations (2) and (3) would be the polar and azimuthal angles of P, respectively. We now define the quantum dissimilarity, as follows.

**Definition** **8**(Quantum Dissimilarity)**.** *For any two points $P_{1},P_{2}\in R^{3}$, we define the **quantum dissimilarity** as*
(43)$$d_{q}\left(P_{1},P_{2}\right):=\frac{1}{2}\left(1-\mathrm{Tr}\left(\psi_{P_{1}}\psi_{P_{2}}\right)\right),$$*where $\psi_{P}$ is the Bloch embedding of P.*

Notice that, as per this definition, the classical two-dimensional points are embedded in pure states only. In Section 3.4, we consider Bloch embedding to be the centroids into mixed states as well, showing that this does not provide an advantage in our framework. This quantum dissimilarity can be obtained either with the SWAP test or with the Bell state measurement on $\psi_{P_{1}}⊗\psi_{P_{2}}$ as described in Section 2.3. In our application, we use the Bell state measurement (depicted in Figure 2), as we do not need the extra resources of the SWAP test that allow us to keep the post-measurement state. For more details, see Lemma 1.

By Equation (5), the quantum dissimilarity is proportional to the cosine dissimilarity (this might not be true for other definitions of quantum dissimilarity, as in Section 3.4 where we redefine it to include embedding into mixed states)
(44)$$d_{q}\left(P_{1},P_{2}\right)=\frac{1}{2}\left(1-\mathrm{Tr}\left(\psi_{P_{1}}\psi_{P_{2}}\right)\right)=\frac{1}{4}\left(1-\frac{P_{1}}{∥P_{1}∥}\cdot \frac{P_{2}}{∥P_{2}∥}\right)=\frac{1}{4}d_{s}\left(P_{1},P_{2}\right)$$It is also proportional to the Euclidean distance for points on the same sphere (points with the same magnitude) as per Lemma 2. Namely, if $s_{1},s_{2}\in S^{2}\left(r\right)$ then:(45)$$\begin{array}{cc} d_{q}\left(s_{1},s_{2}\right) & =\frac{1}{8r^{2}}d_{e}\left(s_{1},s_{2}\right). \end{array}$$

We can finally define stereographic embedding as follows.

**Definition** **9**(Stereographic Embedding)**.** *We define the **stereographic embedding** of a classical vector $p\in R^{2}$ as*
*1.* *Projecting the 2D point p into a point on the sphere of radius r in a 3D space through the ISP:*(46)$$s:=s_{r}^{-1}\left(p\right)\in S^{2}\left(r\right)\subset R^{3};$$*2.* *Bloch embedding s into $\psi_{s}=\psi_{s_{r}^{-1}\left(p\right)}$.*

Comparing the distance estimate of the Stereographic embedding procedure (Equations (45) and (A18)) with that for the hybrid quantum-classical *k*-means with angle embedding (the ‘distance loss function’ described in [10]), we can observe that the theoretical performance has been improved, since the estimate has been much improved with respect to the closeness to Euclidean distance. This leads us to expect a performance improvement of the SQ-*k*NN algorithm over the hybrid quantum-classical implementation of quantum *k*-means with angle embedding.

A very time-consuming computational step of *k*NN involves the repeated calculations of distances between the dataset points meant to be classified and each centroid. In the case of the quantum *k*NN in [2], since angle embedding is not injective, many steps must be spent after estimating the fidelity to calculate the distance between the points using the norms. Even in [32,33,37], the norms of the points have to be stored classically, leading to much computational expense. Our method has the clear benefit of calculating the cosine dissimilarity directly through fidelity estimation. No further calculations are required due to all stereographically projected points having the same norm *r* in the sphere, and the existence of a bijection between the ISP and the original 2D datapoints, thus saving computational time and resources. In summary, Equations (44) and (45) portray a method to measure a dissimilarity that leads to consistent clustering involving pure states.

As one can observe, in the case of stereographically projected points, $d_{q}$ is directly proportional to the Euclidean dissimilarity between them. Since all the points after the projection into the sphere have equal modulus *r*, and each projected point corresponds to a unique 2D data point, we can *directly compare the probability of obtaining outcome ij=11 on the Bell-state measurement circuit for cluster assignment*. This eliminates extra steps needed during computation to account for the different moduli of points on the two-dimensional plane.

### 3.2. The SQ-kNN Algorithm

We now have all the building blocks to define the quantum clustering algorithm. The quantum part will be the dissimilarity estimation $d_{q}$, obtained by embedding the data into quantum states as described in Section 3.1 and then feeding it into the quantum circuit described in Section 2.3 and Figure 2 to estimate an outcome probability. The finer details of distance estimation are further described in Appendix E. We can now formally define the developed algorithm building on the definition of clustering state (Definition 1), of clustering algorithm and cluster update provided by Definitions 2 and 3, of ISP as defined in Section 2.7, and of quantum dissimilarity $d_{q}$ from Definition 8.

**Definition** **10**(Stereographic Quantum *k*NN (SQ-*k*NN))**.** *Let $s_{r}^{-1}$ be the ISP, let $d_{q}$ be the quantum dissimilarity, and let $\left(D,\bar{c},R^{2},d_{e}\right)$ be a clustering state (where D and $\bar{c}$ are two-dimensional datasets and initial centroids). We then define the **Stereographic Quantum kNN (SQ-kNN)** as the kNN clustering algorithm*
(47)$$\begin{array}{c} \left(s_{r}^{-1}\left(D\right),\;s_{r}^{-1}\left(\bar{c}\right),\;R^{3},\;d_{q},\;{\bar{c}}_{q}^{\mathrm{update}}\right) \end{array}$$*where $c_{q}^{\mathrm{update}}:=\sum_{p\in C}p$.*

The complete process of performing SQ-*k*NN in practice can be described in detail, as follows.

First, prepare to embed the classical data and initial centroids into quantum states using the ISP: project the two-dimensional datapoints and initial centroids (in our case, the alphabet) into a sphere of radius *r*, and calculate the polar and azimuthal angles of the points. This first step is executed entirely on a *classical* computer.Cluster Update: The calculated angles are used to create the states using Bloch embedding (Definition 7). The dissimilarity between the centroid and point is then estimated using the Bell-state measurement. Once the dissimilarities between a point and all the centroids have been obtained, the point is assigned to the cluster of the ‘closest’ centroid. This is repeated for all the points that have to be classified. The quantum circuit and classical controller handle this step entirely. The controller feeds in the classical values at the appropriate times, stores the results of the various shots and classifies the point to the appropriate cluster.Centroid Update: Since any non-zero point on the subspace of $c_{s}$ (see Corollary 1, Figure 4) is an equivalent choice, to minimise the computational expense, the centroids are updated as the sum point of all points in the cluster—as opposed to the average, for example, which minimises the Euclidean dissimilarity (Equation (20)).

Once the centroids are updated, Step 2 (Cluster Update) is repeated, followed once again by Step 3 (Centroid Update) until a decided stopping condition is fulfilled.

Compared to 2D quantum *k*NN clustering with angle or amplitude embedding, the differences with the SQ-*k*NN algorithm lie in the embedding and the post-processing after the inner-product estimation.

The stereographic embedding of the 2D datapoints is conducted by theinverse stereographic projecting the point into a sphere of a chosen radius and then producing the quantum state obtained by rescaling the sphere to radius one.In contrast, in angle embedding, the coefficients of the vectors are used as the angles of the Bloch vector (also known as dense angle embedding [44]), while in amplitude embedding, they are used as the amplitudes in the standard basis. For 2D vectors, amplitude embedding allows one to encode only one coefficient (instead of two) in one qubit, and sometimes angle embedding would also encode only one coefficient by using a single rotation (standard angle embedding [45]). Both angle and amplitude embeddings require the lengths of the vectors to be stored classically beside the quantum state, which is not needed in Bloch embedding.No post-processing is needed after the overlap estimation of stereographically embedded data, as the obtained estimate is already a linear function of the inner product, as opposed to standard approaches using angle or amplitude encoding. Amplitude embedding also requires non-trivial computational time in the state preparation process. In contrast, in angle embedding, though the state preparation time is constant, recovering a useful dissimilarity (e.g., Euclidean) may involve many post-processing steps.


*In short, stereographic embedding has the advantage of angle over amplitude embedding of being able to encode all values of a vector and low state preparation time, and the advantage of amplitude versus angle embedding in the recovery of the dissimilarity.*


### 3.3. Complexity Analysis and Scaling

Let the ISP of a *d*-dimensional point $p=[x_{1}\;x_{2}\;\ldots \;x_{d}]$ into $S^{d}\left(r\right)$, using the point (r,0,0,…,0) (the ‘North pole’) as the projection point, be the point $s=[s_{0}\;s_{1}\;\ldots \;s_{d}]$. It is known that the Cartesian coordinates of s are given by:(48)$$\begin{array}{cccc} s_{0} & =r\frac{{∥p∥}^{2}-r^{2}}{{∥p∥}^{2}+r^{2}}\; & s_{i} & =\frac{2r^{2}x_{i}}{{∥p∥}^{2}+r^{2}}\;\forall \;i=1,\cdots ,d \end{array}$$
where ${∥p∥}^{2}=\sum_{j=1}^{d}x_{j}^{2}$. Hence one can observe that the time complexity of the projection for a single d-dimensional point is O(d), provided we only need the Cartesian coordinates. However, for the Stereographic Embedding procedure, one would need to calculate the angles made by s with the axes of the space, making the time complexity of projection O(poly(d)). Therefore, the total time complexity of the Stereographic Embedding for a d-dimensional dataset *D* of size |D|=N and *k* centroids is given by O((k+N)poly(d)). We now specify two strategies for scaling our algorithm for higher dimensional datapoints.

#### 3.3.1. Using Qubit-Based System

We consider the case where we have two *d*-dimensional vectors $p_{1},p_{2}\in R^{d}$, and we want to compute the quantum dissimilarity of vectors. If we have a qubit-based system and use dense angle encoding for encoding the stereographically projected point, we would encode the *d* calculated angles using $\frac{d}{2}$ qubits. Namely, for the (d+1)-dimensional projection of a *d*-dimensional point $p_{1}$, one would obtain *d* angles $\left[\theta_{1}\;\theta_{2}\;\ldots \;\theta_{d}\right]$ that specify the projected point $s_{1}$ on $S^{d}\left(r\right)$. We then encode this vector using the same unitary, as follows:(49)$$\begin{array}{c} |\psi_{1}\rangle =\underset{j\in (1,3,\ldots ,d-1)}{⨂}|\psi_{1_{j}}\rangle =\underset{j\in odd(d)}{⨂}U\left(\theta_{j},\theta_{j+1},0\right)|0\rangle . \end{array}$$If *d* is odd, we can pad $\left[\theta_{1}\;\theta_{2}\;\ldots \;\theta_{d}\right]$ with an extra 0 to make it even. The other point $s_{2}=s_{r}^{-1}\left(p_{2}\right)$ will be encoded into the state
(50)$$\begin{array}{c} |\psi_{2}\rangle =\underset{j^{\prime }\in \left(1,3,\ldots ,d-1\right)}{⨂}|\psi_{2_{j^{\prime }}}\rangle =\underset{j^{\prime }\in odd\left(d\right)}{⨂}U\left(\theta_{j^{\prime }}^{\prime },\theta_{j^{\prime }+1}^{\prime },0\right)|0\rangle . \end{array}$$Now, to find the overlap between the states, one would have to perform the Bell-state measurement (Section 2.3) pairwise using $|\psi_{1_{j}}\rangle$ and $|\psi_{2_{j^{\prime }}}\rangle$ as inputs, i.e.,
(51)$$\begin{array}{c} {\left|\langle \phi_{11}||\psi_{1_{j}}\rangle ⊗|\psi_{2_{j^{\prime }}}\rangle \right|}^{2}=\frac{1}{2}{\left(1-|\langle \psi_{1_{j}}|\psi_{2_{j^{\prime }}}\rangle \right|}^{2}) \end{array}$$In the common practical case of the vectors being expressed in an orthogonal basis, one would only have to find the overlap for $j=j^{\prime }$. We would then have the quantum dissimilarity by adding up the individual probabilities
(52)$$\begin{array}{c} d_{q}\left(p_{1},p_{2}\right)=\sum_{j\in odd(d)}{\left|\langle \phi_{11}||\psi_{1_{j}}\rangle ⊗|\psi_{2_{j}}\rangle \right|}^{2} \end{array}$$This procedure has a time complexity of O(d). It is important point to note that this quantum dissimilarity will no longer correspond directly to the inner product or Euclidean distance between either $p_{1}$ and $p_{2}$ or $s_{1}$ and $s_{2}$. With the strategy of pairwise overlap estimation, we observe that if the number of shots to estimate $|\langle \phi_{11}|(|\psi_{1_{j}}\rangle ⊗|\psi_{2_{j}}\rangle {|}^{2}$ is kept constant, the error in estimation will be ∝d. Hence, taking into account the increase in the number of shots to estimate the quantum dissimilarity with a given total error ϵ, the time complexity of this qubit implementation of overlap estimation between two points using SQ-*k*NN scales is $O(\epsilon^{-1}poly\left(d\right))$. Hence, for all points and clusters, the time complexity would be $O(\epsilon^{-1}kNpoly\left(d\right))$.

It is shown in [46] that collective measurements are a better strategy than repeated individual measurements for overlap estimation. Although this is shown in [46] for estimating overlap between two states given the availability of multiple copies of the same states, similar collective measurement strategies could be applied in this case for better results. In conclusion, the time complexity of SQ-*k*NN for qubit-based implementation is
(53)$$O(\epsilon^{-1}kNpoly\left(d\right))$$


#### 3.3.2. Using Qudit-Based System

Consider
$$|1\rangle :=\sum_{i\in \left\{\;0,\ldots ,d-1\;\right\}}|ii\rangle .$$Then, for any two real vectors $|\psi \rangle =\sum \psi_{i}|i\rangle$ and $|\phi \rangle =\sum \phi_{i}|i\rangle$, namely when $\psi_{i},\;\phi_{i}\in R$, we have
$$\begin{array}{c} \langle 1|\left(|\psi \rangle ⊗|\phi \rangle \right)=\sum \psi_{i}\phi_{i}=\langle \phi^{*}|\psi \rangle =\langle \phi |\psi \rangle . \end{array}$$Now, we make a qudit Bell measurement, which can be obtained as in Figure 2 by replacing the Hadamard with the Fourier transform and the qubit CNOT with the qudit CNOT$\left(\sum \left|i\rangle \langle i\right|⊗\left|i+j\;\mathrm{mod}\;d\rangle \langle j\right|\right)$ (if we have multiple qubits instead of qudits, then the solution is even simpler: perform a qubit Bell measurement with each pair of qubits because the tensor product of maximally entangled states is still a maximally entangled state). Then, one of the basis states of this von Neumann measurement will be
$$|\Phi \rangle \equiv {|\Phi \rangle }_{d}=\frac{1}{\sqrt{d}}|1\rangle .$$Thus, the inner product between two real vectors can still be measured with a Bell measurement, but the resulting probability of measuring outcome |Φ〉 scales as
$${\left|\langle \Phi |\left(|\psi \rangle ⊗|\phi \rangle \right)\right|}^{2}=\frac{1}{d}{\left|\langle \phi |\psi \rangle \right|}^{2};$$
meaning that, as the inner product remains constant going to higher dimensions, the number of shots needed to estimate the inner product with constant precision scales polynomially in the dimension. In contrast, such complexity for the SWAP test remains constant because the contribution of the fidelity to the outcome probability is not divided by the dimension *d*. This is why the SWAP test is usually considered for inner product estimation, even if in the case of qubits the Bell measurement is a simpler solution [25].

### 3.4. SQ-kNN and Mixed States

Instead of estimating the quantum dissimilarity, we can use the datapoints produced by the ISP to perform classical *k*NN clustering on the 3D-projected data. We called this the 3DSC-*k*NN (3D Stereographic Classical *k*NN) in Definition 6. This algorithm produces centroids that are inside the sphere. As previously pointed out, when computing the Euclidean 3D centroid on the data projected on the sphere, the result is a point inside the sphere rather than on the sphere itself.

In the Bloch sphere, internal points are mixed states, namely states with added noise. In contrast, the quantum algorithm (SQ-*k*NN) always produces pure centroids, namely points on the surface of the sphere. The only noiseless states are the pure states on the surface of the sphere, and thus the intuition is that arguably mixed states should not help. However, this is not immediately clear from the algorithm. Comparing 3DEC-*k*NN to SQ-*k*NN, it is thus natural to ask whether embedding the centroids into mixed states inside the Bloch sphere improves the accuracy.

Here, we show that the intuition is correct, namely that projecting into the pure state centroid is a better option. The reason is that while the quantum dissimilarity is proportional to the Euclidean dissimilarity for states in the same sphere, the same is not true for Bloch vectors with different lengths.

To allow for mixed state embedding, we can modify the definition of quantum dissimilarity (Equation (45)) to produce mixed states whenever the 3D vector has a length of less than one. This results in the following new quantum dissimilarity.

**Definition** **11**(Noisy Quantum Dissimilarity)**.** *Let $B^{2}\left(1\right)=\left\{P\in R^{3}\;|\;∥P∥\leq 1\right\}$ be the ball of radius 1. We define the **noisy quantum dissimilarity** as the function ${\tilde{d}}_{q}:B^{2}\left(1\right)\times B^{2}\left(1\right)\to R$*
(54)$$\begin{array}{cc} {\tilde{d}}_{q}\left(P_{1},P_{2}\right):= & \frac{1}{2}\left(1-\mathrm{Tr}(\rho_{P_{1}}\rho_{P_{2}})\right) \end{array}$$*where $\rho_{P}$ is the quantum state of the Bloch vector P as in Equation *(4)*.*

Now suppose we have a convex combination of pure states; namely, suppose we have
(55)$$\begin{array}{cccc} \bar{\rho } & =\sum_{i}p_{i}\rho_{P_{i}}\; & {∥P∥}_{i} & =1\;\forall \;i. \end{array}$$where $p_{i}$ is a probability distribution ($p_{i}>0$ and $\sum p_{i}=1$). By linearity, we have
(56)$$\begin{array}{cccc} \bar{\rho } & =\rho \left(\sum p_{i}P_{i}\right)=:\rho \left(\bar{P}\right)\; & \bar{P} & =\sum p_{i}P_{i}\;, \end{array}$$By convexity, $\bar{P}$ will always lie in the sphere. Namely, we have $∥\bar{P}∥\leq 1$ and thus by linearity
(57)$$\begin{array}{cc} {\tilde{d}}_{q}\left(\bar{P},P\right) & =\sum p_{i}{\tilde{d}}_{q}\left(P_{i},P\right)=\sum p_{i}d_{q}\left(P_{i},P\right). \end{array}$$The result in Equation (57) can be interpreted as another two-step process: first, repeatedly performing the Bell-state measurement of *each state* $\rho_{P_{i}}$ that makes up the cluster and $\rho_{P}$ corresponding to the datapoint, to estimate each individual dissimilarity; and then, taking the weighted average of the dissimilarities according to the composition of the mixed state centroid. This procedure is clearly impractical experimentally and also no longer correlates to the cosine dissimilarity for mixed states.

Computing the diagonalization of $\bar{\rho }$ as per Equation (8)
(58)$$\begin{array}{cccc} \rho & =p\rho \left(\frac{P}{∥P∥}\right)+\left(1-p\right)\rho \left(\frac{-P}{∥P∥}\right)\; & p & =\frac{1}{2}\left(1+∥\bar{P}∥\right)\\ & =p\psi_{\bar{P}}+\left(1-p\right)\psi_{-\bar{P}} \end{array}$$(where ψ is the Bloch embedding) makes the estimation more practical by reducing it to two estimations of $d_{q}$, namely
(59)$$\begin{array}{cc} {\tilde{d}}_{q}\left(\bar{P},P\right) & =p{\tilde{d}}_{q}\left(\frac{\bar{P}}{∥\bar{P}∥},P\right)+\left(1-p\right){\tilde{d}}_{q}\left(-\frac{\bar{P}}{∥\bar{P}∥},P\right)\\ & =pd_{q}\left(\bar{P},P\right)+\left(1-p\right)d_{q}-\left(\bar{P},P\right) \end{array}$$The implementation portrayed at Equation (59) simplifies the measurement procedure of the mixed state. Furthermore, instead of estimating $d_{q}\left(\pm \bar{P},P\right)$ separately, the estimation can be performed directly by preparing ψ(P) with probability *p* and ψ(−P) with probability 1−p, and finally collecting all the outcomes in a single estimation, which requires a larger number of shots to achieve the same precision of estimation. Another issue is that the points $\bar{P},-\bar{P}$ have to be computed, which is quite time-consuming. This is true even for Equation (57); however, a number of shots proportional to the number of Bloch vectors $P_{i}$ in the cluster is needed for an accurate estimation. Regardless, linearity and convexity make it clear that using mixed states can only increase the quantum dissimilarity.

Namely, while in Euclidean dissimilarity, points inside the sphere can reduce the dissimilarity, the quantum dissimilarity is proportional to the Euclidean dissimilarity only for unit vectors and actually increases for points inside the Bloch sphere. Hence, we conclude that the behaviour of 3DSC-*k*NN does not carry over to SQ-*k*NN.

## 4. Quantum-Inspired Stereographic Nearest-Neighbour Clustering (2DSC-kNN)

We have detailed the developed quantum algorithm in the previous Section 3. This section develops the classical analogue to this quantum algorithm—the ‘quantum-inspired’ classical algorithm. A table summarising all the algorithms discussed in this paper, including the next one, can be found in Table 1. We begin by defining this analogous classical algorithm in terms of the clustering state (Definition 1), deriving a relationship between the Euclidean and spherical centroids given datapoints that lie on a sphere, and then proving our claim that the defined classical algorithm and previously described stereographic quantum *k*NN are indeed equivalent.

Recall from Lemma 3 that
(60)$$\begin{array}{cc} c_{s}^{\mathrm{update}}\left(C\right):= & \underset{x\in S^{n}\left(r\right)}{\mathrm{argmin}}\sum_{p\in C}d_{s}\left(x,p\right)=r\frac{\sum_{p\in C}p}{∥\sum_{p\in C}p∥}. \end{array}$$

**Definition** **12**(Two-Dimensional Stereographic Classical *k*NN (2DSC-*k*NN))**.** *Let $s_{r}^{-1}$ be the ISP, and let $\left(D,\bar{c},R^{2},d_{e}\right)$ be a 2D euclidean clustering state. We define the 2D Stereographic Classical kNN (2DSC-kNN) as*
(61)$$\begin{array}{c} \left(s_{r}^{-1}\left(D\right),\;s_{r}^{-1}\left(\bar{c}\right),\;S^{2}\left(r\right),\;d_{s},\;{\bar{c}}_{s}^{\mathrm{update}}\right). \end{array}$$

**Remark** **3.**
*Notice that due to the cluster update being cosine (${\bar{c}}_{s}^{\mathrm{update}}$) and Lemma 3, we can equivalently substitute $d_{s}$ with $d_{e}$, namely we can substitute it without changing the outcome of the cluster update. In our implementation, we use the Euclidean dissimilarity for simplicity of coding.*


To expand upon Definition 12, for the quantum-inspired/classical analogue stereographic *k*NN, the steps of execution are as follows:Stereographically project all the 2-dimensional data and initial centroids into the sphere $S^{2}\left(r\right)$ of radius r. Notice that the initial centroids will lie on the sphere by construction.Cluster Update: Form the clusters using the method defined in Equation (16), i.e., form all $C_{j}\left({\bar{c}}_{i}\right)$. Here, $D=S^{2}\left(r\right)$ and dissimilarity $d=d_{s}\left(p,c\right)=\frac{1}{2r^{2}}d_{e}\left(p,c\right)$ (Definition 5 and Lemma 2).Centroid Update: A closed-form expression for the centroid update was calculated in Equation (35) $\left(c_{s}^{\mathrm{updated}}=\frac{r}{∥\sum_{p\in C}p∥}\sum_{p\in C}p\right)$. This expression recalculates the centroid once the new clusters have been formed. Once all the centroids are updated, Step 2 (cluster update) is then repeated, and so on, until a stopping condition is met.

### 4.1. Equivalence

We now want to show that the 2DSC-*k*NN algorithm of Definition 12 is equivalent to the previously defined quantum algorithm using stereographic embedding (Definition 10). For that, we first define the equivalence of two clustering algorithms.

**Definition** **13**(Equivalence of Clustering Algorithms)**.** *Let $K=(D,{\bar{c}}_{1},D,d,{\bar{c}}_{\mathrm{update}})$ and $K^{\prime }=\left(D^{\prime },{\bar{c}}_{1}^{\prime },D^{\prime },d^{\prime },{\bar{c}}_{\mathrm{update}}^{\prime }\right)$ be two clustering algorithms. They are said to be equivalent if there exists a transformation $t:D\to D^{\prime }$ such that it maps the data, initial centroids and centroid update, and clusters of K to the data, initial centroids and centroid update, and clusters of $K^{\prime }$; namely if*
*1.* *$D^{\prime }=t\left(D\right)$,**2.* *${\bar{c}}_{1}^{\prime }=t\left({\bar{c}}_{1}\right)$ and ${\bar{c}}_{\mathrm{update}}^{\prime }=t\circ {\bar{c}}_{\mathrm{update}}$,**3.* *$C_{j}^{\prime }\left(t\left(\bar{c}\right)\right)=t\left(C_{j}\left(\bar{c}\right)\right)$ for all j=1,…,k and any $\bar{c}\in D^{k}$.**where we apply t elementwise and thus $t\left(\bar{c}\right)=\left(t\left(c_{1}\right),\ldots ,t\left(c_{k}\right)\right)$ for any list of centroids $\bar{c}$ and $t\left(C\right)=\left\{\;t(p):p\in C\;\right\}$ for any set of datapoints C, and where $C_{j}$ are the clusters as defined in Equation (16).*

**Theorem** **1.**
*SQ-kNN (Definition 10) and 2DSC-kNN (Definition 12) are equivalent.*


**Proof.** By definition, let $\left(D,{\bar{c}}_{1},R^{2},d_{e}\right)$ be the 2D clustering state, thus giving us the SQ-*k*NN algorithm as
(62)$$\begin{array}{c} K=\left(S,\;{\bar{s}}_{1},\;R^{3},\;d_{q},\;{\bar{c}}_{q}^{\mathrm{update}}\right) \end{array}$$
and the 2DSC-*k*NN clustering algorithm as
(63)$$\begin{array}{c} K^{\prime }=\left(S,\;{\bar{s}}_{1},\;S^{2}\left(r\right),\;d_{s},\;{\bar{c}}_{s}^{\mathrm{update}}\right) \end{array}$$
where
(64)$$\begin{array}{cccc} S:= & s_{r}^{-1}\left(D\right)\; & {\bar{s}}_{1}:= & s_{r}^{-1}\left({\bar{c}}_{1}\right) \end{array}$$Let us use the notation $\hat{p}:=\frac{p}{∥p∥}$ and define the transform $t:R^{3}\mapsto S^{2}\left(r\right)$ as $t\left(p\right)=r\;\hat{p}$, which rescales any vector to have length *r*. Observe that trivially for all $p\in S^{2}\left(r\right)$, t(p)=p and thus $t\circ s_{r}^{-1}=s_{r}^{-1}$. Therefore
(65)$$\begin{array}{cccc} t(S) & =S,\; & t({\bar{s}}_{1}) & ={\bar{s}}_{1}\;. \end{array}$$Moreover, the equivalence of centroids is obtained since
(66)$$\begin{array}{c} t\left(c_{q}^{\mathrm{update}}\left(C\right)\right)=t\left(\sum_{p\in C}p\right)=r\frac{\sum_{p\in C}p}{∥\sum_{p\in C}p∥}=c_{s}^{\mathrm{update}}\left(C\right). \end{array}$$For the clusters, we prove the equivalence of the cluster updates as follows. We will use $d_{s}\left(a,b\right)=4\cdot d_{q}\left(a,b\right)$ (Equation (44)) and the fact that $d_{s}$ and $d_{q}$ are invariant under *t*, namely $d_{s}\circ t=d_{s}$ and $d_{q}\circ t=d_{q}$, or more explicitly
(67)$$\begin{array}{cccc} d_{s}\left(t\left(a\right),b\right) & =d_{s}\left(a,b\right)=d_{s}\left(a,t\left(b\right)\right),\; & d_{q}\left(t\left(a\right),b\right) & =d_{q}\left(a,b\right)=d_{q}\left(a,t\left(b\right)\right). \end{array}$$Let now $\bar{s}\in {\left(R^{3}\right)}^{k}$. Then, using the above equations and that t(s)=s,t(S)=S, we have
(68)$$\begin{array}{cc} C_{j}^{\prime }\left(t\left(\bar{s}\right)\right) & =\left\{\;p\in S\;|\;d_{s}\left(p,t\left(s_{j}\right)\right)\leq d_{s}\left(p,t\left(s_{\ell }\right)\right)\;\forall \;\ell =1,\ldots ,k,\;p\notin \underset{\ell <j}{⋃}C_{\ell }^{\prime }\left(t\left(\bar{s}\right)\right)\;\right\}\\ & =\left\{\;p\in S\;|\;d_{q}\left(p,s_{j}\right)\leq d_{q}\left(p,s_{\ell }\right)\;\forall \;\ell =1,\ldots ,k,\;p\notin \underset{\ell <j}{⋃}C_{\ell }\left(\bar{s}\right)\;\right\} \end{array}$$where the change in the dissimilarity inequality has also transformed the calculation of $C_{\ell }^{\prime }\left(\bar{s}\right)$ into the calculation of $C_{\ell }\left(\bar{s}\right)$. We are now finished, since t(s)=s for s∈S and thus
(69)$$\begin{array}{cc} C_{j}^{\prime }\left(t\left(\bar{s}\right)\right) & =\left\{\;t(p)\in t(S)\;|\;d_{q}\left(p,s_{j}\right)\leq d_{q}\left(p,s_{\ell }\right)\;\forall \;\ell =1,\ldots ,k,\;p\notin \underset{\ell <j}{⋃}C_{\ell }\left(\bar{s}\right)\;\right\}\\ & =(C_{j}\left(\bar{s}\right)) \end{array}$$This concludes the proof    □

The following discussion provides a visual intuition of Theorem 1. In Figure 4, the sphere with centre origin (O) and radius *r* is the stereographic sphere into which the two-dimensional points are projected, while the sphere with centre O and radius 1 is the Bloch sphere. The points $p_{1},p_{2},\ldots ,p_{n}$ are the stereographically projected points defining a cluster, corresponding to the previously used labels $s_{1},s_{2},\ldots ,s_{n}$. The centroid $c_{e}$ is obtained with the euclidean average in $R^{3}$. In contrast, the centroid $c_{s}$ is restricted to be in $S^{2}\left(r\right)$ and equal $c_{e}$ rescaled to lie on this sphere. The quantum states $|\psi_{p_{1}}\rangle ,|\psi_{p_{2}}\rangle ,\ldots ,|\psi_{p_{n}}\rangle$ are obtained after Bloch embedding the stereographically projected points $p_{1},p_{2},\ldots ,p_{n}$, and $|\psi_{c}\rangle$ is the quantum state obtained after Bloch embedding the centroid. The points marked on the Bloch sphere in Figure 4 are the Bloch vectors of the quantum states $|\psi_{p_{1}}\rangle ,|\psi_{p_{2}}\rangle ,\ldots ,|\psi_{p_{n}}\rangle$ and $|\psi_{c}\rangle$.

One can observe from Definition 7 that the origin, any point p on the sphere, and $|\psi_{p}\rangle$, are collinear. Hence, it can be observed that in the process of SQ-*k*NN clustering, the points on the stereographic sphere are projected radially into the sphere of radius 1. Once the labels were assigned in the previous iteration, the new centroid is computed, giving an integer multiple of the average point $c_{e}$, which lies within the stereographic sphere. Crucially, when we embed this new centroid into the quantum state for the quantum dissimilarity calculation of the next step, since we only use the polar and azimuthal angle of the point for embedding (see Definition 7), *the prepared quantum state is also projected into the surface of the Bloch sphere*—or, in other words, a pure state is prepared ($|\psi_{c}\rangle$). Hence, we can observe that all the dissimilarity calculations in the SQ-*k*NN will take place between points on the *surface* of the Bloch sphere, even though the calculated quantum centroid is *contained outside* the stereographic sphere. This argument also illustrates why *any* point on the ray $\vec{Oc_{e}c_{s}}$ can be used for the centroid update step of the stereographic quantum *k*NN; any chosen point on the ray, when embedded into a quantum state for dissimilarity calculations will reduce to $|\psi_{c}\rangle$.

In short, we know from Lemma 3 that $O,c_{e},$ and $c_{s}$ lie on a straight line. Therefore, one can observe that if the Bloch sphere is scaled by r, the point on the Bloch sphere corresponding to $|\psi_{c}\rangle$ will transform to $c_{s}$, i.e., $0,|\psi_{c}\rangle ,c_{e}$ and $c_{s}$ are all collinear. Equation (45) shows that SQ-*k*NN clustering clusters points on a sphere as per Euclidean dissimilarity; that implies that simply scaling the sphere makes no difference to the clustering. Therefore, we conclude that clustering on the surface of the stereographic sphere $S^{2}$ (2DSC-*k*NN) is equivalent to the quantum algorithm with stereographic embedding (SQ-*k*NN).

### 4.2. Complexity Analysis and Scaling

As we showed in Section 3.2, the time complexity of ISP for calculating Cartesian coordinates of a d-dimensional vector is O(d). Hence, the total time complexity of projection for the 2DSC-*k*NN will be O((k+N)d), where N=|D| is the total number of points, and *k* is the total number of centroids. Since the cluster update step uses Euclidean dissimilarity, it will take O(kNd) time in total (O(d) for each distance calculation, which is to be conducted for each pair of *N* points and *k* centroids). The centroid update expression (Equation (35)) can be calculated in O(Nd), making the total time for this step O(kNd) since we have *k* centroids. Hence, we have
(70)$$\begin{array}{c} \mathrm{Time}\;\mathrm{complexity}\;\mathrm{of}\;2\mathrm{DSC}-k\mathrm{NN}\;\mathrm{algorithm}=O(kNd), \end{array}$$
on par with the classical *k*-means clustering algorithm, and at least polynomially faster than the stereographic quantum *k*NN (Equation (53)) or Lloyd’s quantum clustering algorithm (taking into account input assumptions) [2,4,9].

## 5. Experiments and Results

We defined the procedure for SQ-*k*NN in Section 3. Section 3.1 introduces our idea for state preparation—projecting the two-dimensional data points into a *higher* dimension. Section 3.1 details the hybrid quantum-classical method used for our process and then proves that the output of the quantum circuit is not only a valid but also an excellent metric that can be used for distance estimation between two points. Section 4 describes the quantum-inspired classical algorithm analogous to the quantum algorithm (2DSC-*k*NN). In this section, we test and compare the quantum-inspired rather the quantum algorithm for two main reasons:The hardware performance and availability of quantum computers (NISQ devices) is currently so much worse than that of classical computers that no advantage can likely be obtained with the quantum algorithm.The goal of this paper is not to show a “quantum advantage” in time complexity over the classical *k*-means in the NISQ context—it is to show that stereographic projection can lead to better learning for classical clustering and be a better embedding for quantum clustering. In particular, the equivalence between 2DSC-*k*NN and SQ-*k*NN proves that noise is the only limitation for the stereographic quantum algorithm to achieve the accuracy of the quantum-inspired algorithm.

All the experiments were carried out on a server with the following specifications: 2 Intel Xeon E5-2687W v4 chips clocked at 3.0 GHz (24 cores/48 threads), 128 GB RAM. All experiments are performed on the real-world 64-QAM data provided by Huawei (see Section 2.1 and Appendix A). Due to the extensive nature of testing and the large volume of analysis generated, we do not present all the figures in the following sections. Figures which sufficiently demonstrate general trends and observations have been included here. An exhaustive collection of all figures and other such analysis results, as well as the source code, real-world input data, and collected output data, can be accessed at [47].

The terminology used is as follows:Radius: the radius of the stereographic sphere into which the two-dimensional points are projected.Number of points: the number of points upon which the clustering algorithm was performed. For every experiment, the selected points were a random subset of all the 64-QAM data (of a specific noise) with cardinality equal to the required number of points. The random subset is created using the numpy.random.sample() from the Python Numpy library.Number of runs: Since for each choice of parameters for each experiment we select a subset of points at random, we repeat each of the experiments many times to remove bias from the random choice and obtain stable averages and standard deviations for the collected performance parameters (described in another list below). This number of repetitions is the “number of runs”.Dataset Noise: As explained in Section 2.1, data were collected for four different input powers. Data are divided into four datasets labelled with powers 2.7, 6.6, 8.6, and 10.7 dBm.Natural endpoint: The natural endpoint of a clustering algorithm occurs when
(71)$$\begin{array}{cccc} C_{j}\left({\bar{c}}_{i+1}\right) & =C_{j}\left({\bar{c}}_{i}\right) & \forall & \;j=1,\ldots ,k \end{array}$$
i.e., when all the clusters remained unchanged (stay the same) even after the centroid update. It is the natural endpoint since if the clusters do not change, the centroids will not change either in the next iteration, leading to the same clusters (Equation (71)) and centroids for all future iterations.

The algorithms that we test are:2DSC-*k*NN: The quantum-analogue algorithm of Definition 12, the classical equivalent of the SQ-*k*NN and the most important candidate for our testing.2DEC-*k*NN: The standard classical *k*NN of Definition 4 implemented upon the original 2-dimensional dataset (n=2), which serves as a baseline for performance comparison.3DSC-*k*NN: The standard classical *k*NN, but implemented upon the stereographically projected 2-dimensional dataset, as defined in Definition 6. We again emphasise that in contrast to the 2DSC-*k*NN, the centroid lies *within* the sphere, and in contrast to the 2DEC-*k*NN, the clustering takes place in $R^{3}$. This algorithm serves as another control, to gauge the relative impacts of stereographically projecting the dataset versus restricting the centroid to the surface of the sphere. It is an intermediate step between the 2DSC-*k*NN and the 2DEC-*k*NN algorithms.

From these algorithms, we measure the following performance parameters (or KPIs, Key Performance Indicators):Accuracy: Since we have the true labels of the datapoints available, we can measure the accuracy of the algorithm as the percentage of points that have been given the correct label, i.e., symbol accuracy rate. All accuracies are recorded as a percentage.Symbol or Bit error rate: As mentioned in Appendix A, due to Gray encoding, the bit error rate is approximately $\frac{1}{6}$ of the symbol error rate, which in turn is simply one minus the accuracy. Although error rates are the standard performance parameter in channel coding, we decided to measure the accuracy instead, which is the standard performance parameter in machine learning.Accuracy gain: The gain is calculated as *(accuracy of candidate algorithm minus accuracy of two-dimensional classical k-means clustering algorithm)*, i.e., it is the increase in accuracy of the algorithm over the baseline, defined as the accuracy of the classical *k*-means clustering algorithm acting on the 2D dataset for those number of points.Number of iterations: One iteration of the clustering algorithm occurs when the algorithm performs the cluster update followed by the centroid update (the algorithm must then perform the cluster update again). The number of times the algorithm repeats these two steps before stopping is the number of iterations. We use the number of iterations the algorithm requires to reach its ‘natural endpoint’ as a proxy for *convergence performance*. The lesser the number of iterations performed, the faster the algorithm’s convergence. The number of iterations does not directly correspond to time performance since the time taken for one iteration differs between all algorithms.Iteration gain: The gain in iterations is defined as *(the number of iterations of 2D k-means clustering algorithm minus the number of iterations of candidate algorithm)*, i.e., the gain is how many fewer iterations the candidate algorithm took than the 2DEC-*k*NN algorithm to reach its natural endpoint.Execution time: The amount of time taken for a clustering algorithm to provide the final output (the final centroids and clusters) given the two-dimensional data points as input, i.e., the time taken end to end for the clustering process. All times in this work are recorded in *milliseconds* (ms).Execution time gain: This gain is calculated as *(the execution time of 2DEC-kNN k-means clustering algorithm minus the execution time of candidate algorithm)*.Overfitting Parameter: The difference in testing and training accuracy.

With these algorithms and variables, we perform two main experiments:The Overfitting Test: The dataset is divided into a ‘training’ and a ‘testing’ set, to characterise the clustering and classification performance of the algorithms.The Stopping Criterion Test: The iterations and other performance parameters are varied, to test whether and what kind of stopping criterion is required.

We observe that the tested algorithms display some very promising and interesting results. We manage to obtain improvements in accuracy and convergence performance almost across the board, and we discover the very important optimisation parameters of the radius of projection and the stopping criterion.

### 5.1. Experiment 1: Overfitting

Here, the datasets were divided into training and testing data. First, a random subset of cardinality equal to the number of points was chosen from the dataset, and then 80% of the selected points were assigned as ‘Training Data’, while the other 20% was assigned as ‘Testing Data’.

In the training phase, the algorithms were first run on the training data with the maximum possible iterations set to 50 to keep an acceptable running time. The stopping criterion for all algorithms was chosen as the natural endpoint—the algorithm stopped either when the number of iterations hit 50, or when the natural endpoint was reached (whichever happened first). The final centroid co-ordinates (${\bar{c}}_{\mathrm{last}\;\mathrm{iteration}}$) were recorded in the training phase, to be used for the testing phase, along with several performance parameters. The recorded performance parameters were the algorithm’s accuracy, the number of iterations taken, and the execution time.

Once the training was over, the centroids calculated at the end of training were used as the initial centroids for the testing set datapoints, and the algorithm was run with the maximum number of iterations set to 1, i.e., the calculated centroids were then used to classify the remaining points as per the dissimilarity and dataspace of each algorithm. The recorded performance parameters were the algorithm’s accuracy and execution time. Once both the testing and training accuracy had been recorded, the overfitting parameter was also recorded.

For each set of input variables (just the number of points for 2DEC-*k*NN clustering, the radius and number of points for the 2DSC-*k*NN and 3DSC-*k*NN clustering), the entire experiment (training and testing) was repeated 10,000 times in batches of 100 to calculate reasonable standard deviations for every performance parameter.

There are several reasons for this choice of experiment:It exhaustively covers all the parameters that can be used to quantify the performance of the algorithms. We were able to observe very important trends in the performance parameters with respect to the choice of radius and the effect of the number of points (affecting the choice of when one should trigger the clustering process on the collected received points).It avoids the commonly known problem of overfitting. Though this approach is not usually used in testing the *k*NN due to its iterative nature, we felt that from a machine learning perspective, it is useful to know how well the algorithms perform in a classification setting as well.Another reason that justifies the training and testing approach (clustering and classification) is the nature of the real-world application setup. When transmitting QAM data through optical-fibre, the receiver receives only one point at a time and has to classify the received point to a given cluster in real-time using the current centroid values. Once a number of data points have accumulated, the *k*NN algorithm can be run to update the centroid values; after the update, the receiver will once again perform classification until some number of points has been accumulated. Hence, we can observe that in this scenario, the clustering and the classification performance of the chosen method become important.

#### 5.1.1. Results

We begin the presentation of the results of this experiment by first showing the characterisation of the 2DSC-*k*NN algorithm with respect to the input variables.

Figure 5 characterises the testing and training accuracy of the 2DSC-*k*NN algorithm acting upon the 2.7 dBm dataset, i.e., classification and clustering performance, respectively. Figure 6 portrays the same results in the form of a heat map, with a focus on the region of interest of the algorithm. These figures are representative of the trends of all four datasets.

Figure 7 characterises the convergence performance of the quantum algorithm—it shows how the number of iterations required to reach the natural endpoint of the 2DSC-*k*NN algorithm varies as the number of points and radius of projection changes. Once again, the figures for all the other datasets follow the same pattern as the included figures.

We then compare the performance of the 2DSC-*k*NN algorithm with that of the 3DSC-*k*NN and 2DEC-*k*NN algorithms.

Accuracy Performance: Here, in all the following figures, the winner is chosen as the radius for which the maximum accuracy is achieved for the given number of points. Figure 8 depicts the variation in testing accuracy with the number of points for all three algorithms along with error bars. As mentioned before, this characterises the performance of the algorithms in a ‘classification’ mode, that is, when the received points must be decoded in real-time. Figure 9 portrays the trend in training accuracy with the number of points for all three algorithms along with error bars. This characterises the performance of the algorithms in ‘clustering’ mode, that is, when the received points must be used to update the centroid for future re-classification or if the received datapoints are stored and decoded in batches. Figure 8 and Figure 9 also plot the gain in testing and training accuracies respectively for the 3DSC-*k*NN and 2DSC-*k*NN algorithms. The label of the points in these figures is the radius of ISP for which that accuracy gain was achieved.Iteration Performance: Here, in all the following figures, the winner is chosen as the radius for which the minimum number of iterations is achieved for the given number of points. Figure 10 shows how the required number of iterations for all three algorithms varies as the number of points increases. Figure 10 also displays the gain of the 2DSC-*k*NN and 3DSC-*k*NN algorithms in the number of iterations to reach their natural endpoints. The label of the points in these figures is the radius of ISP for achieving that iteration gain.Time Performance: Here, in all the following figures, the winner is chosen as the radius for which the minimum execution time is achieved for the given number of points. Figure 11 puts forth the dependence of testing execution time upon the number of points for all three algorithms along with error bars. As mentioned before, these times are effectively the amount of time the algorithm takes for one iteration. This characterises the performance of the algorithms when performing direct classification decoding of the received points in real time.This figure reveals the trend in training execution time with the number of points for all three algorithms, along with error bars. This characterises the time performance of the algorithms when performing clustering—that is, when the received points must be used to update the centroid for future re-classification or if the received datapoints are stored and decoded in batches. Figure 12 plots the gains in testing and training execution times for the 3DSC-*k*NN and 2DSC-*k*NN algorithms.Overfitting Performance: Figure 13 exhibits how the overfitting parameter for the 2DEC-*k*NN *k*-means clustering, 3DSC-*k*NN, and 2DSC-*k*NN algorithms vary as the number of points changes.

#### 5.1.2. Discussion and Analysis

From Figure 5, we can observe that there is an *ideal radius* > 1 for which maximum accuracy is achieved. This ideal radius is usually between two and five for our datasets. For a good choice of radius (>1), the accuracy increases monotonically (with an upper bound) with the number of points. In contrast, for a poor choice of radius (<1), the accuracy nosedives as the number of points increases. This is due to the clusters getting squished together near the North pole of the stereographic sphere (the point (0,0,r)). If one is dealing with a large number of points, the accuracy becomes even more sensitive to the choice of radius, as the decline in accuracy for a bad radius is much steeper as the number of points increases. These observations hold for both training and testing accuracy (classification and clustering), regardless of the noise in the dataset. These observations are also well-reflected in the heatmaps, where one can observe that the best training and testing performance is for r=2 to 3 and the maximum number of points. It would seem that choosing too large of a radius is not too harmful. This might hold true for the classical algorithms, but when the quantum algorithm is deployed, all the points will be clustered around the South pole of the Bloch sphere and even minimal noise in the quantum circuit will degrade performance. Hence, there is a sweet spot of the radius to be chosen.

Figure 7 also shows that there is an ideal radius > 1 for which one needs the minimum number of iterations to reach the natural endpoint. This ideal radius is once again between two and five for our datasets. As the number of points increases, the number of iterations always increases. The increase is minimal for a good choice of radius, while for a bad choice, the convergence is very slow. For our experiments, we chose the maximum iterations as 50, hence the observed plateau is at 50 iterations. If one is dealing with a large number of points, the convergence becomes more sensitive to the choice of radius. The increase in iterations for a poor choice of radius is much steeper. 2DSC-*k*NN algorithm and 3DSC-*k*NN algorithm display near-identical performance.

From Figure 8, we can observe that both 2DSC-*k*NN algorithm and 3DSC-*k*NN algorithm perform better in accuracy than the 2DEC-*k*NN algorithm for all datasets. The advantage becomes more definitive as the number of points increases, as the increase in accuracy moves beyond the error bar. We observe the highest increase in accuracy for the 2.7 dBm dataset.

In Figure 9, one can observe the noticeably better performance of the 2DSC-*k*NN algorithm and 3DSC-*k*NN algorithm over the 2DEC-*k*NN algorithm for all datasets than in the testing case (classification mode). Once again, the 2.7 dBm dataset shows the maximum increase. The advantage again becomes more definitive as the number of points increases as the increase in accuracy moves beyond the error bar. The 2DSC-*k*NN algorithm and 3DSC-*k*NN algorithm show an almost identical performance.

From Figure 8 and Figure 9, we can also observe that almost universally for both algorithms, the gain is greater than 0, i.e., we beat the 2DEC-*k*NN algorithm in nearly every case. We can also observe that the best radius is almost always between two and five. Another observation is that the gain in training accuracy increases with the number of points. The figures further display how similarly the 3DSC-*k*NN algorithm and 2DSC-*k*NN algorithm perform in terms of accuracy, regardless of noise.

From Figure 10, it can be concluded that for low noise datasets, since the number of iterations is already quite low, there is not much gain or loss; all three algorithms perform almost identically. For high-noise datasets, however, both the 3DSC-*k*NN algorithm and 2DSC-*k*NN algorithm show significant performance improvement, especially for a higher number of points. For a high number of points, the improvement is beyond the error bars and hence very significant. It can be noticed that the ideal radius for minimum iterations is once again between two and five. Here, also, the 3DSC-*k*NN algorithm and 2DSC-*k*NN algorithm perform similarly, with the 2DSC-*k*NN algorithm performing better in certain cases.

One learns from Figure 11 that most importantly, the 2DSC-*k*NN algorithm and 2DEC-*k*NN algorithm take nearly the same amount of time for execution in classification mode, and the 2DSC-*k*NN algorithm in most cases beats the 3DSC-*k*NN algorithm. Here too, the gain is significant, since it is much beyond the error bar. The execution time increases linearly with the number of points, as expected. These conclusions are supported by Figure 12. Since the 2DSC-*k*NN algorithm takes almost the same time, and provides greater accuracy, *it is an ideal candidate to replace the 2DEC-kNN algorithm for classification applications.*

Figure 11 also shows that all three algorithms take almost the same amount of time for training, i.e., in clustering mode. The 3DSC-*k*NN and 2DSC-*k*NN algorithms once again perform almost identically, almost always slightly worse than 2DEC-*k*NN clustering. Figure 12 supports these observations. Here, execution time increases linearly with the number of points as well, as expected. In Figure 13, all three algorithms have nearly identical performance. As expected, the overfitting decreases with an increase in the number of points.

### 5.2. Experiment 2: Stopping Criterion

Based on the results obtained from the first experiment, we performed another experiment to see how the accuracy of the algorithms varies iteration by iteration. It was observed that the natural endpoint of the algorithm was rarely the ideal endpoint in terms of performance. Hence, we wished to observe the performance of each algorithm as the number of iterations progressed.

In this experiment, the entire random subset of datapoints was used for the clustering algorithm. The algorithms were run on the dataset, and the accuracy of the algorithms at each iteration as well as the iteration number of the natural endpoint was recorded. The maximum number of iterations was once again 50. By repeating this 100 times for each number of points (and radius, if applicable), we obtained the general performance variation of each algorithm with the iteration number. The input variables were the number of points, the radius of the stereographic sphere and the iteration number; the recorded performance parameters were the accuracy and probability of stopping.

This experiment revealed that the natural endpoint was indeed a poor choice of stopping criterion and that the endpoint should be chosen per some “loss function”. It also revealed some important trends in the performance parameters which not only emphasised the importance of the choice of radius and number of points but also provided greater insight into the disadvantages and advantages of each algorithm.

#### 5.2.1. Results

Characterisation of the 2DSC-*k*NN algorithm: Figure 14 depicts the dependence of the accuracy of the 2DSC-*k*NN algorithm upon the iteration number and projection radius for the 2.7 dBm dataset. The figures for the rest of the datasets follow the same trends and are nearly identical in shape.Figure 15 shows the dependence of the probability of the 2DSC-*k*NN algorithm reaching its natural endpoint versus the radius of projection and iteration number for the 10.7 dBm dataset with 51,200 points and for the 2.7 dBm dataset with 640 points. Once again, the figures for the rest of the datasets follow the same trends and their shape can be extrapolated from the presented Figure 15.Comparison with 2DEC-*k*NN and 3DSC-*k*NN Clustering: Figure 16 portrays the gain of the 2DSC-*k*NN and 3DSC-*k*NN algorithms in the number of iterations to reach maximum accuracy for the 2.7 and 10.7 dBm datasets. In these figures, a gain of ‘*g*’ means that the algorithm took ‘*g*’ fewer iterations than the classical *k*-means acting upon the 2D dataset did to reach maximum accuracy.Figure 17 plots the gain of the 2DSC-*k*NN and 3DSC-*k*NN algorithms in the maximum achieved accuracy for the 2.7 and 10.7 dBm datasets. Here, a gain of ‘*g*’ means that the algorithm was g% more accurate than the maximum accuracy of the classical *k*-means acting upon the 2D dataset.Lastly, Figure 18 illustrates the maximum accuracies achieved by the 2DSC-*k*NN, 3DSC-*k*NN, and 2DEC-*k*NN algorithms for the 2.7 and 10.7 dBm datasets.

#### 5.2.2. Discussion and Analysis

Figure 14 shows that once again, there is an ideal radius for which maximum accuracy is achieved. The ideal projection radius is larger than one; in particular, it seems to be between two and five. Most importantly, there is *an ideal number of iterations for maximum accuracy, beyond which the accuracy reduces.* As the number of points increases, the sensitivity of the accuracy to radius increases significantly. For a bad choice of radius, accuracy only falls with an increase in the number of iterations and stabilises at a very low value. For a good radius, accuracy increases to a point as iterations proceed, and then stabilises at a slightly lower value. If the allowed number of iterations is restricted, the choice of radius to achieve the best results becomes extremely important. With a good radius one can achieve nearly the maximum possible accuracy with very few iterations. As mentioned before, this holds for all dataset noises. As the dataset noise increases, the iteration number at which the maximum accuracy is achieved also expectedly increases. Since accuracy always falls after a point, choosing a stopping criterion is essential rather than waiting for the algorithm to reach its natural endpoint. An idea for the stopping criterion is to record the sum of the average dissimilarity for each centroid at each iteration and stop the algorithm if that quantity increases.

Figure 15 portrays that for a good choice of radius, the 2DSC-*k*NN algorithm approaches convergence much faster. For r<1, the algorithm converges much slower or never converges. As the number of points increases, the convergence rate for the poor radius falls dramatically. For a radius greater than the ideal radius as well, the convergence rate is lower. As one would expect, the algorithm takes longer to converge as the dataset noise increases. As mentioned before, if the number of iterations is severely limited, the choice of radius becomes very important. The algorithm can reach its ideal endpoint in very few iterations if the radius is chosen well.

Through Figure 16, we observe that for lower values of noise, both algorithms do not produce much advantage in terms of iteration gain, regardless of the number of points in the dataset. However, both algorithms significantly outperform the classical one at higher noise in the dataset and a high number of points. This effect is especially significant for the 2DSC-*k*NN algorithm. For the highest noise and all the points, it saves over 20 iterations compared to the 2DEC-*k*NN algorithm—*an advantage of over 50%*. One of the reasons for this is that at low noises, the algorithms already perform quite well, and it is at high noise with a high number of points that the algorithm is stressed enough to reveal the difference in performance. It should be noted that these gains are much higher than when the algorithms are allowed to reach their natural endpoint, suggesting another reason for choosing an ideal stopping criterion.

Figure 17 shows that for all datasets and numbers of points, the two algorithms perform better than 2DEC-*k*NN clustering. The 3DSC-*k*NN algorithm and 2DSC-*k*NN algorithms perform nearly the same, and the accuracy gain seems to stabilise with an increase in the number of points. Figure 18 supports these conclusions.

### 5.3. Overall Observations

#### 5.3.1. Overall observations from Experiment 1:

The ideal projection radius is greater than one and between two and five. At this ideal radius, one achieves maximum testing and training accuracy, and minimum iterations.In general, the accuracy performance is the same for 3DSC-*k*NN and 2DSC-*k*NN algorithms—this shows a significant contribution of the ISP to the advantage as opposed to ‘quantumness’. This is a significant distinction, not made by any previous work.The 2DSC-*k*NN and 3DSC-*k*NN algorithms lead to an increase in the accuracy performance in general, with the increase most pronounced for the 2.7 dBm dataset.The 2DSC-*k*NN algorithm and 3DSC-*k*NN algorithm provide more iteration performance gain (fewer iterations required than 2DEC-*k*NN) for high noise datasets and for a large number of points.Generally, increasing the number of points favours the 2DSC-*k*NN and 3DSC-*k*NN algorithms, with the caveat that a good radius must be carefully chosen.

#### 5.3.2. Overall observations from Experiment 2:

These results further stress the importance of choosing a good radius (two to five in this application) and a better stopping criterion. The natural endpoint is not suitable.The results justify the fact that the developed 2DSC-*k*NN algorithm has significant advantages over 2DEC-*k*NN *k*-means clustering and 3DSC-*k*NN clustering.The 2DSC-*k*NN algorithm performs nearly the same as the 3DSC-*k*NN algorithm in terms of accuracy, but for iterations to achieve this max accuracy, the 2DSC-*k*NN algorithm is better (especially for high noise and a high number of points).The developed 2DSC-*k*NN algorithm and 3DSC-*k*NN algorithm are better than the 2DEC-*k*NN algorithm in general—in terms of accuracy and iterations to reach that maximum accuracy.The supremacy of the 2DSC-*k*NN algorithm over the 2DEC-*k*NN algorithm implies that a fully quantum SQ-*k*NN algorithm would have an advantage over the fully quantum *k*-means algorithm of [2].

## 6. Conclusions and Further Work

This work considers the practical case of performing *k*NN on experimentally acquired 64-QAM data. This work described the problem in detail and explained how the SQ-*k*NN and its classical analogue, the 2DSC-*k*NN clustering algorithm, can be used. The proposed processes and circuits, as well as the theoretical justification for the SQ-*k*NN quantum algorithm and the 2DSC-*k*NN classical algorithm, were described in detail. Finally, the simulation results on the real-world datasets were presented, along with a relevant analysis. From the analysis, one can observe that the classical analogue of the stereographic quantum *k*NN, the 2DSC-*k*NN algorithm, is something that should be considered for industrial implementation—the experiments provide a proof of concept. It also shows the importance of choosing the projection radius and provides a very useful embedding for quantum machine learning algorithms—the generalised stereographic embedding. The theoretical advantage offered by the SQ-*k*NN algorithm over the hybrid quantum-classical quantum *k*-means algorithm was also demonstrated. Another important inference from the obtained results is that the SQ-*k*NN algorithm offers a way to achieve the same advantage compared to the fully quantum *k*-means that 2DSC-*k*NN has over 2DEC-*k*NN—by using the stereographically projected quantum states. These results warrant the practical implementation and testing of both quantum and classical algorithms.

Quantum and quantum-inspired computing has the potential to change the way certain algorithms are performed, with potentially significant advantages. However, as the field is still in relative infancy, finding where quantum and quantum-inspired computing fits in practice is a challenging problem. Here, we have observed that quantum and quantum-inspired computing can indeed be applied to signal-processing scenarios and could potentially work well in the noisy quantum era as clustering algorithms that are relatively robust to noise and inaccuracy.

### Future Work

One of the most important directions of future work is to experiment with more diverse datasets. More experimentation may also lead to more sophisticated methods of selecting the radius for ISP. A more detailed analysis of how to choose a radius of the projection through analytical methods is another important direction for future work. A differential geometric analysis of the effects of the ISP on a square grid provides a rough intuition of why one needs an appropriate radius. A comparison with amplitude embedding is also warranted. The *ellipsoidal projection* (Appendix D) is another promising and novel idea that is to be explored further. In this project, two different stopping criteria for the algorithm were proposed and revealed a change in its performance; yet there is plenty of room to explore more possible stopping criteria.

Further directions of study include improved overlap estimation methods [46] and communication scenarios where the dimensionality of the data points is greatly increased. For example, this happens when multiple carriers experience identical or at least systematically correlated phase rotations.

Another future work is to benchmark against sampling-based quantum-inspired algorithms. As part of a research analysis to evaluate the best possibilities for achieving a practical speed-up, we investigated the landscape of classical algorithms inspired by the sampling in quantum algorithms. Initially, we found that such algorithms have a theoretical complexity competing with quantum algorithms; however, only under arguably unrealistic assumptions on the structure of the classical data. As the performance of the quantum algorithms turns out to be extremely poor, this reopens the possibility that quantum-inspired algorithms can yield performance improvements while we wait for quantum computers with sufficiently low noise. Thus future work will also be a practical implementation of the quantum-inspired *k*NN [4], with the goal of testing the computational advantage over 2DEC-*k*NN, 3DSC-*k*NN, and 2DSC-*k*NN algorithms.

## Figures and Tables

**Figure 1 entropy-25-01361-f001:**
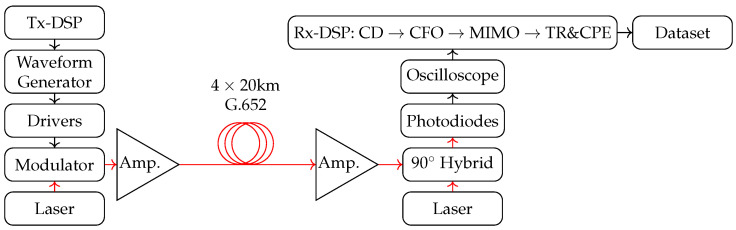
Experimental setup over a 80 km G.652 fibre link at optimal launch power of 6.6 dBm. Chromatic disperion (CD) and carrier frequency offset (CFO) compensation, multiple-input multiple-output (MIMO) equalizer, timing recovery (TR) and carrier phase estimation (CPE) [10,14]. The red arrows distinguish the path of the laser from electrical signals.

**Figure 2 entropy-25-01361-f002:**
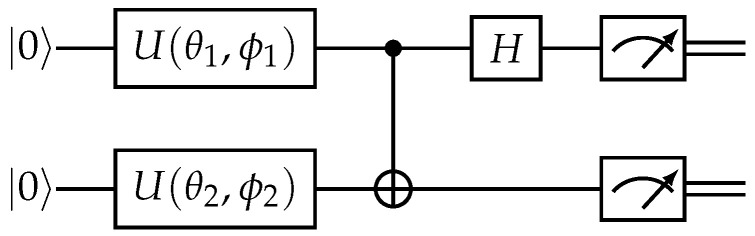
Quantum circuit of the Bell-state measurement. The measurement is obtained by first transforming the Bell basis into the standard basis with (H⊗1)CNOT and then measuring in the standard basis.

**Figure 3 entropy-25-01361-f003:**
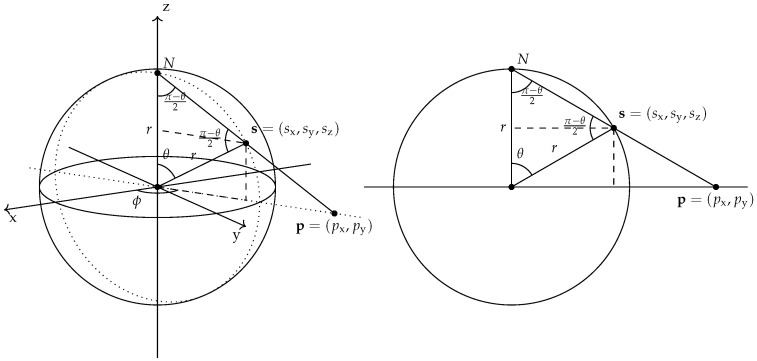
Inverse Stereographic Projection (ISP) with radius *r*. The figure on the right is the cut of the figure in the left going through the *N*, p and the origin.

**Figure 4 entropy-25-01361-f004:**
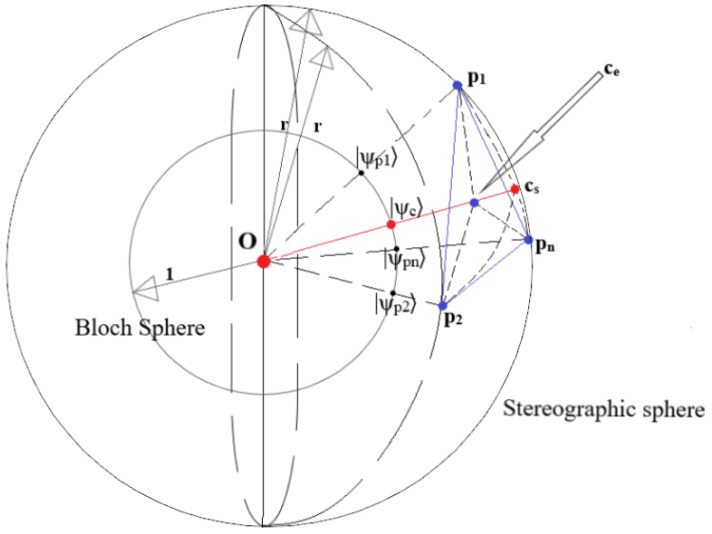
A diagram providing a visual intuition for how the stereographic quantum *k*NN (SQ-*k*NN) is equivalent to the 2DSC-*k*NN. The blue points are the projections of the 2-dimensional datapoints and their corresponding Euclidean centroid, the red points are their corresponding spherical centroid and the Bloch of its quantum state.

**Figure 5 entropy-25-01361-f005:**
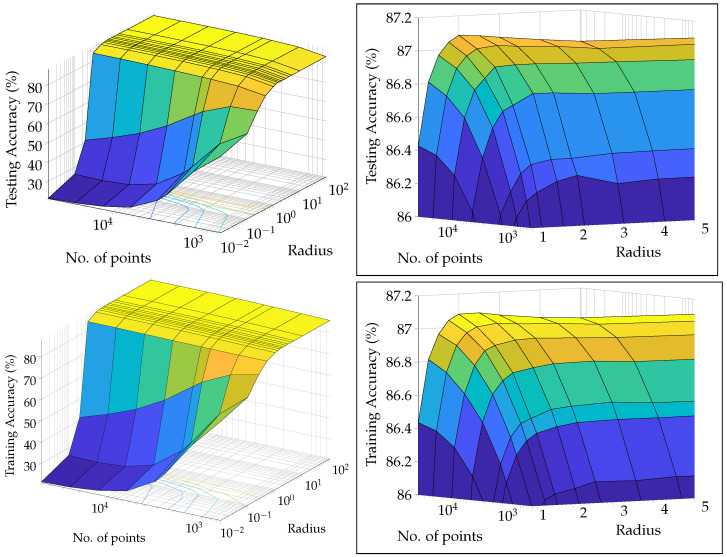
Mean accuracy vs. Number of points vs. Projection radius for the 2DSC-*k*NN algorithm acting upon the 2.7 dBm dataset-Testing (**top left**), close up of testing (**top right**), Training (**bottom left**), close up of training (**bottom right**).

**Figure 6 entropy-25-01361-f006:**
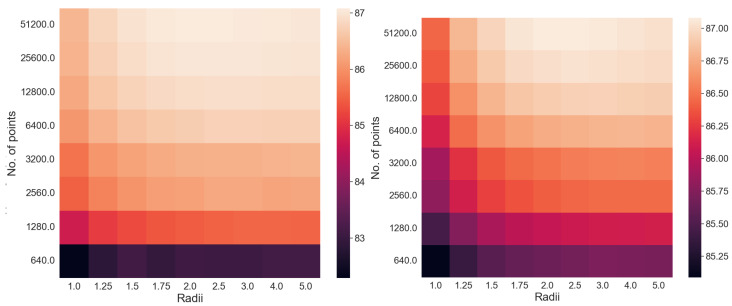
Heat map of Mean accuracy (%) vs. Number of points vs. Projection radius for the 2DSC-*k*NN algorithm acting upon the 2.7 dBm dataset. Testing on the (**left**) and training on the (**right**).

**Figure 7 entropy-25-01361-f007:**
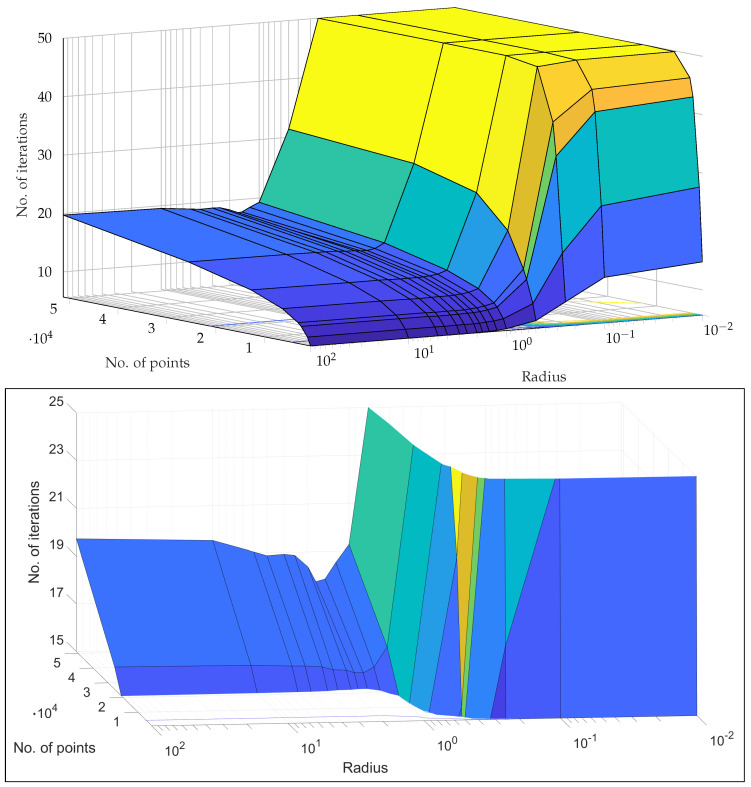
Mean number of *iterations* in training vs. Number of points vs. projection radius for the 2DSC-*k*NN algorithm acting upon the 10.7 dBm dataset. Full data (**top**), close-up (**bottom**).

**Figure 8 entropy-25-01361-f008:**
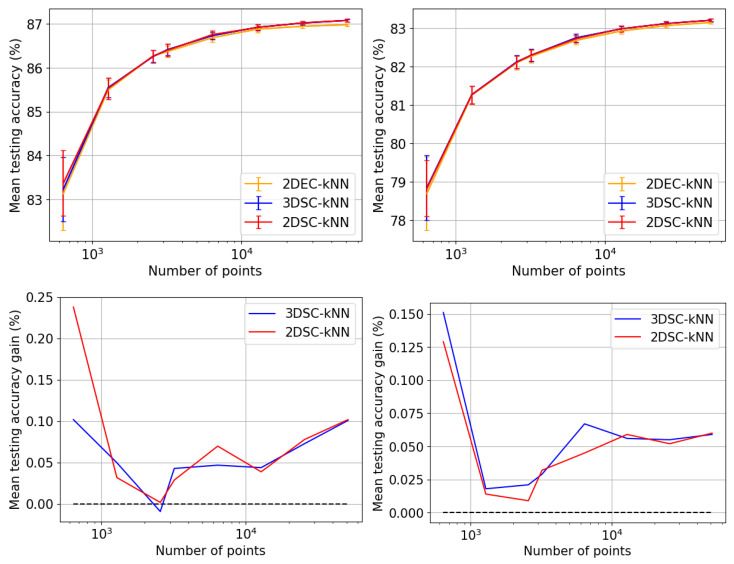
Maximum mean *testing* accuracy and *testing* accuracy *gain* among all tested radii vs. number of points-accuracy for 2.7 dBm (**top left**), accuracy for 10.7 dBm (**top right**), accuracy gain for 2.7 dBm (**bottom left**), accuracy gain for 10.7 dBm (**bottom right**).

**Figure 9 entropy-25-01361-f009:**
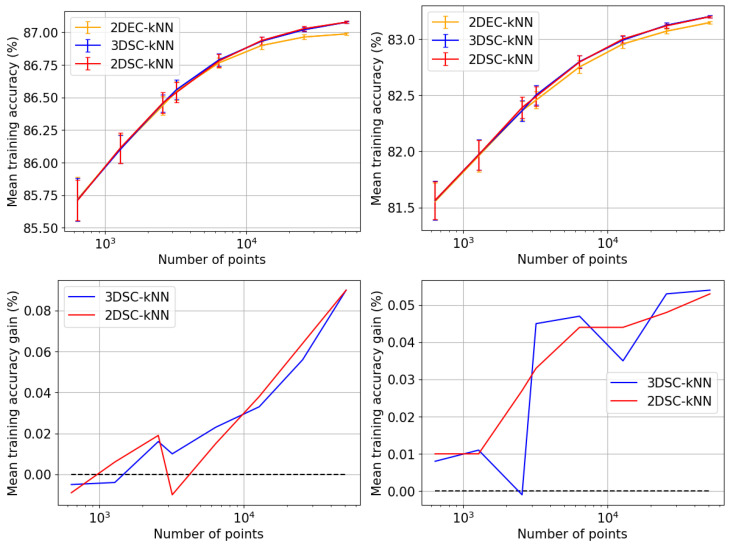
Maximum mean *training* accuracy and *training* accuracy *gain* among all tested radii vs. number of points-accuracy for 2.7 dBm (**top left**), accuracy for 10.7 dBm (**top right**), accuracy gain for 2.7 dBm (**bottom left**), accuracy gain for 10.7 dBm (**bottom right**).

**Figure 10 entropy-25-01361-f010:**
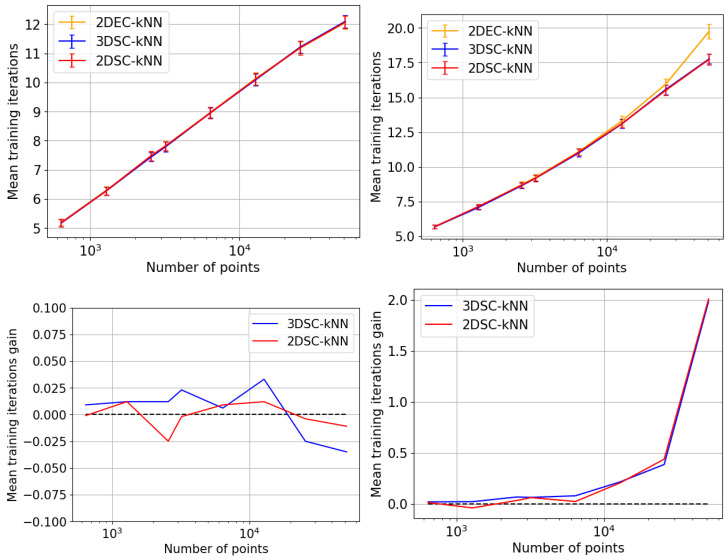
Mean training iterations and iteration gain among all tested radii vs. number of points-iterations for 2.7 dBm (**top left**), iterations for 10.7 dBm (**top right**), iteration gain for 2.7 dBm (**bottom left**), iteration gain for 10.7 dBm (**bottom right**).

**Figure 11 entropy-25-01361-f011:**
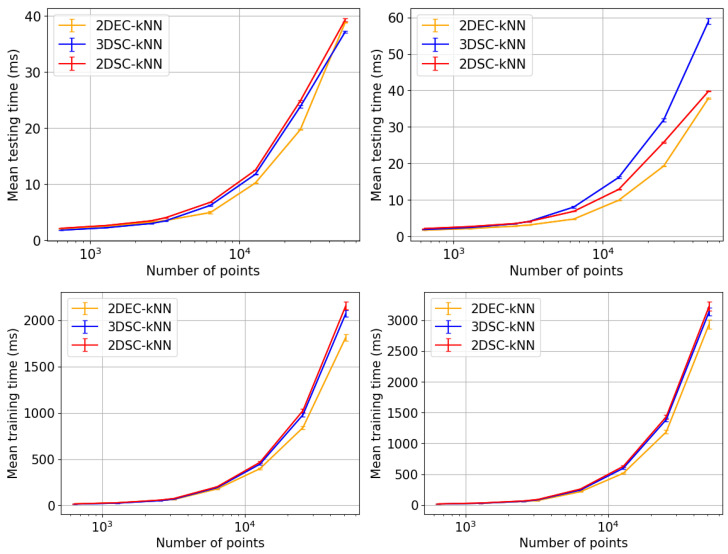
Best mean execution time among all tested radii vs. number of points-testing time for 2.7 dBm (**top left**), testing time for 10.7 dBm (**top right**), training time for 2.7 dBm (**bottom left**), training time for 10.7 dBm (**bottom right**).

**Figure 12 entropy-25-01361-f012:**
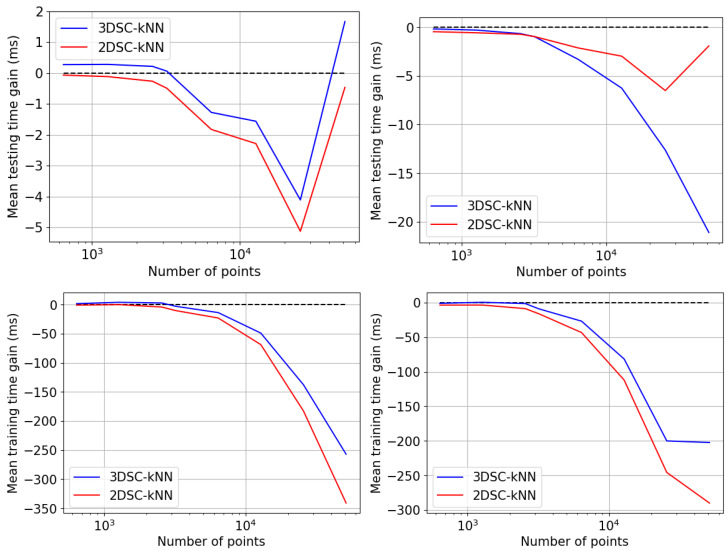
Best mean execution time gain among all tested radii vs. number of points-testing time gain for 2.7 dBm (**top left**), testing time gain for 10.7 dBm (**top right**), training time gain for 2.7 dBm (**bottom left**), training time gain for 10.7 dBm (**bottom right**).

**Figure 13 entropy-25-01361-f013:**
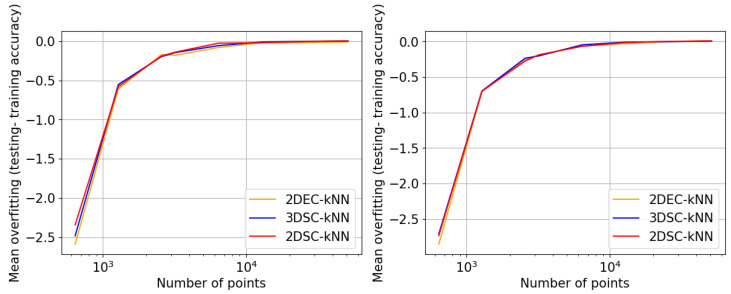
Mean overfitting parameter vs. number of points for the 2.7 dBm (**left**) and 10.7 dBm (**right**) datasets.

**Figure 14 entropy-25-01361-f014:**
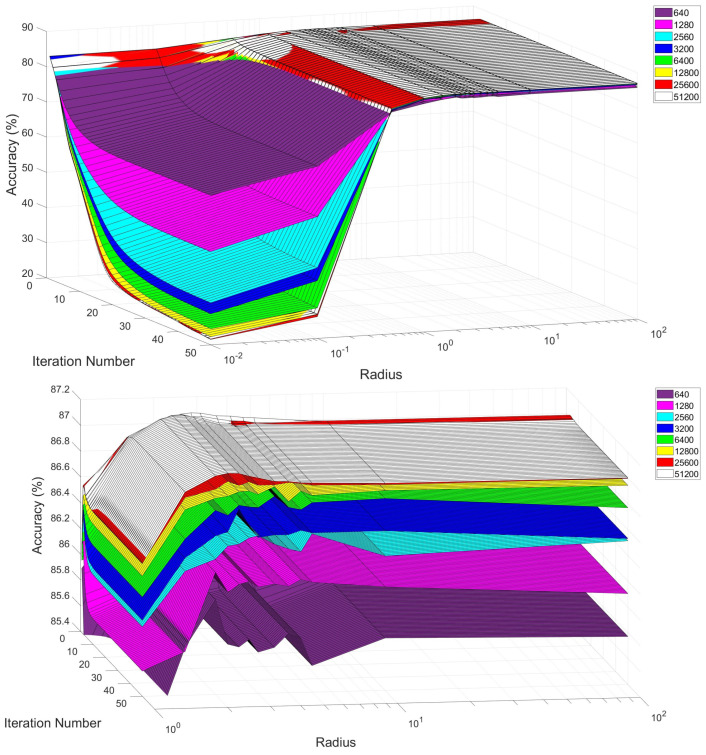
Maximum Accuracy vs. iteration number vs. Projection radius for the 2DSC-*k*NN algorithm acting upon the 2.7 dBm dataset. Close-up of the maximas at the bottom.

**Figure 15 entropy-25-01361-f015:**
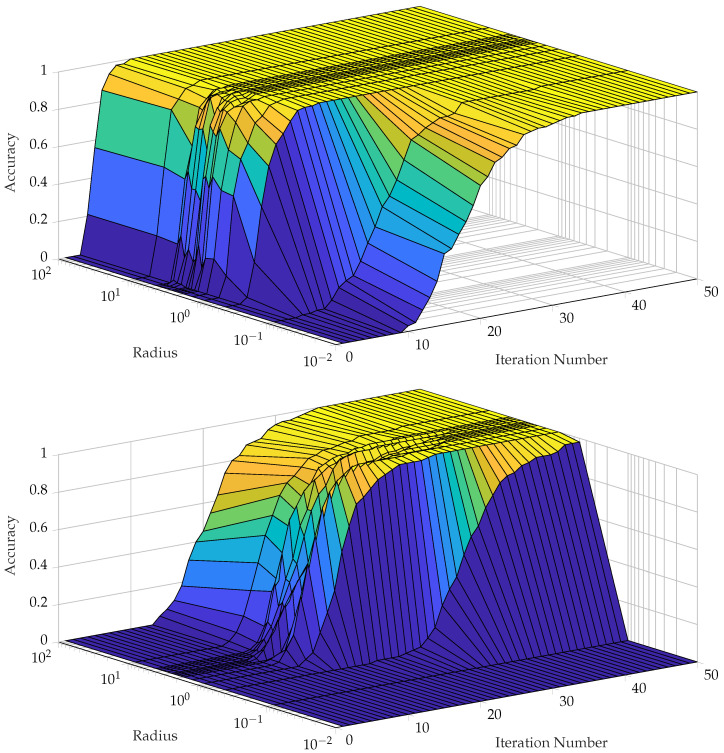
The probability of stopping vs. projection radius vs. iteration number for 2DSC-*k*NN algorithm. (**Top**) 2.7 dBm dataset with the number of points = 640. (**Bottom**) 10.7 dBm dataset with the number of points = 51,200.

**Figure 16 entropy-25-01361-f016:**
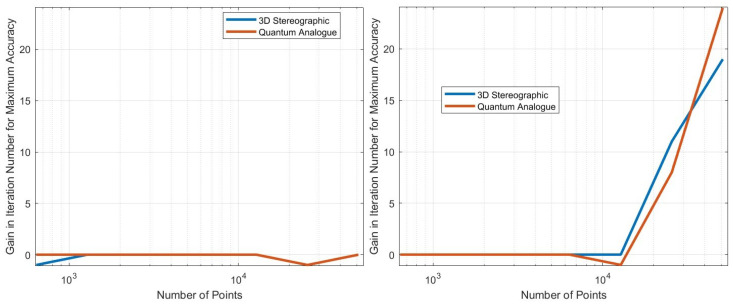
Gain in iteration number at maximum accuracy (number of iterations at maximum accuracy of 2DEC-*k*NN minus the number of iterations at maximum accuracy of 3DSC-*k*NN (blue) and 2DSC-*k*NN (red)) vs. number of points for the 2.7 dBm (**left**) and 10.7 dBm (**right**) datasets.

**Figure 17 entropy-25-01361-f017:**
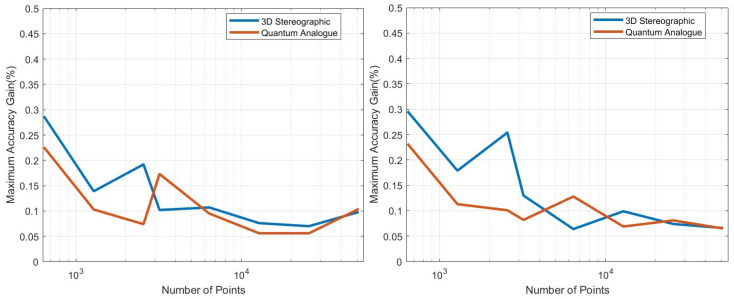
Gain in maximum accuracy of 2DSC-*k*NN (red) and 3DSC-*k*NN (blue) algorithms vs. number of points for the 2.7 dBm (**left**) and 10.7 dBm (**right**) datasets.

**Figure 18 entropy-25-01361-f018:**
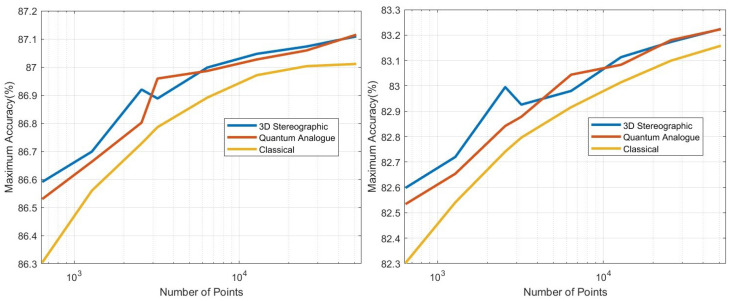
Maximum accuracy of 2DSC-*k*NN (red), 3DSC-*k*NN (blue) and 2DEC-*k*NN (yellow) algorithms vs. number of points for the 2.7 dBm (**left**) and 10.7 dBm (**right**) datasets.

**Table 1 entropy-25-01361-t001:** Summary of various *k*NN algorithms, where SC stands for “stereographic classical” (2D or 3D) and SQ for “stereographic quantum”. Here, *D* is the two-dimensional dataset, $\bar{c}$ are the 2-dimensional initial centroids (initial transmission points), $s_{r}^{-1}$ is the ISP into $S^{2}\left(r\right)$, and the dissimilarities $d_{e}$, $d_{s}$ and $d_{q}$ are defined in Equation (19) and Definitions 5 and 8, respectively. The option of using $d_{e}$ instead of $d_{s}$ in the 2DSC-*k*NN is due to Remark 3.

Algorithm	Reference	Dataset	Initial Centroids	Dataspace	Dissimilarity	Centroid Update
		D	${\bar{c}}_{1}$	D	d	$c^{\mathrm{update}}$
2DEC-*k*NN	Definition 4	*D*	$\bar{c}$	$R^{2}$	$d_{e}$	$\frac{1}{\left|C\right|}\sum_{p\in C}p$
3DSC-*k*NN	Definition 6	$s_{r}^{-1}\left(D\right)$	$s_{r}^{-1}\left(\bar{c}\right)$	$R^{3}$	$d_{e}$	$\frac{1}{\left|C\right|}\sum_{p\in C}p$
2DSC-*k*NN	Definition 12	$s_{r}^{-1}\left(D\right)$	$s_{r}^{-1}\left(\bar{c}\right)$	$S^{2}\left(r\right)$	$d_{s}$ (or $d_{e}$)	$r\frac{\sum_{p\in C}p}{∥\sum_{p\in C}p∥}$
SQ-*k*NN	Definition 10	$s_{r}^{-1}\left(D\right)$	$s_{r}^{-1}\left(\bar{c}\right)$	$R^{3}$	$d_{q}$	$\sum_{p\in C}p$

## Data Availability

All source codes and the complete set of generated graphs are available through [47]. The datasets are published and are a property of Huawei Technologies.

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
