# Peer review of "Quantum and Quantum-Inspired Stereographic K Nearest-Neighbour Clustering"

_entropy, 2023, doi:10.3390/e25091361_

Round 1

Reviewer 1 Report

Dear Authors

The paper presents a very interested topic. I started to read it with great interesting. However, it is not written properly as a journal paper, too long with a lot repeated information, and I had very difficulty time to read it. Please read my comments in the attached file, and rewrite it completely. 

Best!

let's get the paper properly written first, then worry about the English.

Author Response

Dear reviewer, 

Thank you for your extensive reviews. 

Please find attached our reply to all your comments. The paper has undergone extensive revision for better organisation and clarity as per your recommendations. Thanking you,

Yours sincerely, 

Authors

Reviewer 2 Report

In the manuscript, Jasso et.al. proposed a quantum k nearest-neighbor clustering algorithm with stereographic projection for data embedding and a quantum-inspired classical k nearest-neighbor clustering algorithm. The performance and verification of the proposed algorithms were then benchmarked using a real QAM dataset.

I find the proposed methods are clearly conveyed and the paper is well-organized. While I don’t think there is much ‘quantum advantage’ inside, the algorithms using stereographic projection are interesting themselves and could find use cases such as the QAM data the authors showed.  The methods would therefore be inspiring for signal processing problems and for designing novel data embedding techniques.

To this end, I think the paper might meet the requirement for publication upon minor revisions. One concern I have is about the generalization of the proposed inverse stereographic projection into dataset beyond R^2. For high-dimensional cases, what would be cost of finding the projection and designing the unitary operation to map the data? Discussions along this line (at least qualitatively) should be included. Moreover, a detailed analysis on the computation complexity for the projection process and the whole k-NN algorithms that the authors proposed should be given.

Author Response

Dear reviewer, 

Thank you for the interesting and helpful reviews. We have added the sections as per your recommendations. You can find attached a complete response. Thanking you, 

Yours sincerely, 

Authors

Round 2

Reviewer 1 Report

The paper still needs to be improved!

Author Response

Dear Reviewer, 

Thank you for your many helpful suggestions and reviews. We have attached the responses to your questions. We hope that this suffices. Thanking you,

Yours sincerely,

Authors

Reviewer 2 Report

The authors have addressed the previous concerns and the manuscript is more organized than before, even though it could still be simplified by a considerable amount. Also, the figures could be significantly improve. For example, axis labels and legends of many figures are too small to see. 

The authors should perform a throughout check on the manuscript to improve the language and correct remaining grammar issues. 

Author Response

Dear Reviewer, 

Thank you for your helpful suggestions and reviews. We have attached the responses to your questions. We hope that this suffices. Thanking you,

Yours sincerely,

Authors
